# Hi-Patch: Hierarchical Patch GNN for Irregular Multivariate Time Series

Yicheng Luo [1]  Bowen Zhang [1]  Zhen Liu [1]  Qianli Ma [† 1]

## Abstract

Multi-scale information is crucial for multivariate time series modeling. However, most existing time series multi-scale analysis methods treat all variables in the same manner, making them unsuitable for Irregular Multivariate Time Series (IMTS), where variables have distinct origin scales/sampling rates. To fill this gap, we propose Hi-Patch, a hierarchical patch graph network. Hi-Patch encodes each observation as a node, represents and captures local temporal and inter-variable dependencies of densely sampled variables through an intra-patch graph layer, and obtains patch-level nodes through aggregation. These nodes are then updated and re-aggregated through a stack of inter-patch graph layers, where several scale-specific graph networks progressively extract more global temporal and inter-variable features of both sparsely and densely sampled variables under specific scales. The output of the last layer is fed into task-specific decoders to adapt to different downstream tasks. Experiments on 8 datasets demonstrate that Hi-Patch outperforms state-of-the-art models in IMTS forecasting and classification tasks. Code is available at: https://github.com/qianlima-lab/Hi-Patch.

## 1. Introduction

Time series analysis has important applications in various fields such as healthcare (Lan et al., 2024), climate forecasting (Verma et al., 2024) and traffic planning (Li et al., 2024b). Due to the complexity and non-stationarity of real-world systems, multivariate time series often exhibit different variations and fluctuations at different scales (Chen et al., 2024a). Previous studies (Cai et al., 2024; Wang et al., 2024a) have demonstrated that effectively capturing multi-scale features is essential for time series modeling.

Due to sensor malfunctions, varying sampling sources, or human factors, the final sampled time series often exhibits irregularities in real-world applications, resulting in an Irregular Multivariate Time Series (IMTS). The peculiarity of IMTS lies in two aspects: firstly, the sampling interval within each variable is uneven. Secondly, different variables are asynchronously sampled and have distinct sampling rates. The subgraph in the top left corner of Figure 1 illustrates an IMTS example with two variables.

As a unique type of multivariate time series, IMTS also exhibits the inherent multi-scale characteristics of time series. Unfortunately, most existing multi-scale analysis methods assume that the input series are regularly sampled and have limitations in handling IMTS. The mainstream approach of these methods is to downsample or segment the original series based on several fixed time steps (Challu et al., 2023; Shabani et al., 2023), thereby forming several views that reflect features at different scale levels, as shown in the bottom row of Figure 1. On this basis, global temporal and inter-variable dependencies are extracted from coarse-grained views, while local dependencies are extracted from fine-grained views. This approach assumes that all variables have a consistent original scale so that each downsampled view consistently reflects the features of all variables at the same granularity level, thereby achieving consistent separation of multi-scale patterns for all variables.

However, in IMTS, some variables have only a few observations and do not possess fine-grained features, making their downsampled view meaningless (e.g., the variable $V_0$ in Figure 1). Some other variables are densely sampled and require multiple downsampling steps to reflect global patterns (e.g., the variable $V_1$ in Figure 1). In such cases, it is difficult to determine a consistently suitable downsampling level for all variables, making existing methods infeasible from the beginning. Furthermore, in IMTS, a downsampled view can contain mixed granularity features, as shown in the top row of Figure 1, but existing methods tend to extract and analyze single-level features at a specific scale, which are difficult to handle mixed features. Additionally, despite the distinct scales/sampling rates occurring among variables in IMTS, they are not entirely independent but exhibit correlations (Luo et al., 2024). Specifically, variables in MTS may display different inter-variable correlations at different time scales (Cai et al., 2024). How to extract multi-scale

---

[1]School of Computer Science and Engineering, South China University of Technology, Guangzhou, China. Correspondence to: Qianli Ma <qianlima@scut.edu.cn>.

*Proceedings of the $42^{nd}$ International Conference on Machine Learning*, Vancouver, Canada. PMLR 267, 2025. Copyright 2025 by the author(s).

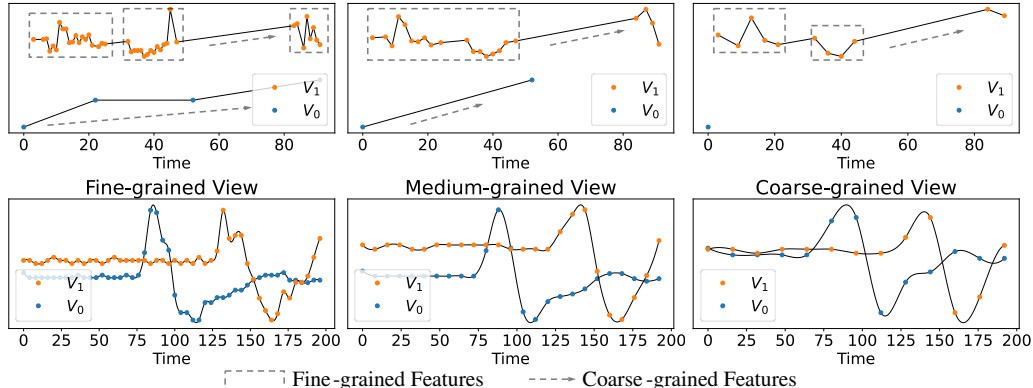

*Figure 1.* Comparison of downsampling results of two variables for an irregular (top row) and a regular (bottom row) multivariate time series with step sizes of 1 (left column), 2 (middle column) and 4 (right column). Irregular time series contain mixed feature levels at a specific downsampling scale, while regular time series exhibit a single feature level at a specific downsampling scale.

temporal dependencies while considering inter-variable correlations under different scales in IMTS, where variables have distinct original scales and asynchronous observations, remains a major challenge in IMTS modeling.

To fill this gap, we propose Hi-Patch, a hierarchical patch graph network. Hi-Patch is based on the concept of patching, which has been recently proven to be effective in capturing local temporal dependencies (Nie et al., 2023; Chen et al., 2024a; Zhang et al., 2024). Hi-Patch first divides the original observation set into multiple patches based on a short time span and encodes each observation as a node. Then, an intra-patch graph layer flexibly represents and fully captures local temporal and inter-variable dependencies of densely sampled variables at the original scale by employing a fully connected graph network within each patch. Subsequently, to capture dependencies at a larger scale, the observation nodes are aggregated to obtain patch-level feature nodes as inputs for several inter-patch graph layers. Each inter-patch graph layer receives specific patch-level nodes, which are obtained through node aggregation of the previous layer, thereby forming a hierarchical architecture. These inter-patch graph layers gradually extract more global temporal and inter-variable dependencies that both densely and sparsely sampled variables possess through scale-specific graph networks. The output of the last layer is finally fed into specific task decoders to adapt to different downstream tasks. During this process, each intra/inter-patch graph layer handles single-level features at a specific scale. Fine-grained features of densely sampled variables are extracted in the lower layers, while coarse-grained features of all variables are extracted in the upper layers, achieving complete extraction of mixed multiple granularities temporal and inter-variable correlations in IMTS. Our main contributions are summarized as follows:

- We introduce the intra-patch/inter-patch graph layers

to flexibly represent and fully extract dependencies of specific variables in IMTS at specific scales.

- We propose Hi-Patch, using a hierarchical architecture to effectively achieve multi-scale modeling of IMTS from fine-grained to coarse-grained.

- We conduct experiments on IMTS forecasting and classification on 8 datasets, and the evaluation results demonstrate that Hi-Patch outperforms existing methods in most cases.

## 2. Related Work

**Irregular Multivariate Time Series Modeling** Existing methods can be generally divided into interpolation-based and raw-data-based methods. The former, employing methods such as kernel-based approaches (Shukla & Marlin, 2019; Wu et al., 2021b), hourly aggregation (Ma et al., 2020) or gaussian process (Tan et al., 2021), aims to regularize sampling intervals. However, interpolation may disrupt the sampling patterns of the original series. Raw-data-based methods learn directly from IMTS. (Che et al., 2018) enhances RNNs for uneven time intervals, while (Horn et al., 2020; Shukla & Marlin, 2021) introduce time embeddings for arbitrary timestamps. (Rubanova et al., 2019; De Brouwer et al., 2019; Biloš et al., 2021; Schirmer et al., 2022; Chen et al., 2024b) use neural ODEs to address irregularities. Recent research also integrates attention mechanisms (Jhin et al., 2021) or graph neural networks (Zhang et al., 2022; Yalavarthi et al., 2024; Zhang et al., 2024; Luo et al., 2024) to capture inter-variable correlations in IMTS. Despite these advancements, most of these methods focus only on single-scale characteristics of IMTS, and how to comprehensively capture multi-scale features within IMTS remains a challenge.

**Multi-scale Modeling for Time Series** Multi-scale information is essential for time series modeling. (Chen et al., 2021) leverages different frequencies to update sub-hidden states, while (Liu et al., 2022) employs pyramid attention for multi-scale feature extraction. (Challu et al., 2023) uses multi-rate sampling and hierarchical interpolation, and (Shabani et al., 2023) assigns forecasting models across temporal scales. (Chen et al., 2024a) introduces a multi-scale Transformer with adaptive paths, and (Wang et al., 2024a) disentangles variations across scales for complementary predictions. However, these methods are designed for regular MTS and are not well adaptive for IMTS. Although Warpformer (Zhang et al., 2023) presents a multi-scale approach for IMTS, its original goal is to balance the differences in sampling densities across variables and it involves upsampling interpolation for sparse variables, which may distort the original data distribution.

**Graph Neural Networks for Multivariate Time Series** Recent studies have integrated GNNs with time series modeling frameworks to effectively capture inter-variable dependencies in MTS (Jin et al., 2024), achieving success in domains such as transportation (Rahmani et al., 2023), healthcare (Wang et al., 2022), and economics (Wang et al., 2021). However, many of these methods are primarily designed for modeling synchronous correlations among variables, lacking sufficient extraction of asynchronous dependencies widely present in IMTS.

## 3. Problem Definition

**Definition 1 (Irregular Multivariate Time Series)** We consider a dataset $\mathcal{D}$ consisting of $n$ IMTS sample. Each sample of $\mathcal{D}$ is a tuple, i.e., $\mathcal{D} := \{(\mathcal{S}_1, y_1), \ldots, (\mathcal{S}_n, y_n)\}$, where $\mathcal{S}_i$ denotes the $i$-th time series sample and $y_i \in \{1, \ldots, C\}$ is the class label ($C$ is the number of categories). We describe $i$-th sample $\mathcal{S}_i$ as a set of $M = |\mathcal{S}_i|$ observations such that $\mathcal{S}_i := \{o_1, \ldots, o_M\}$. Each observation $o_j$ is a tuple $(t_j, z_j, v_j)$, consisting of a timestamp $t_j \in \mathbb{R}^+$, an observed value $z_j \in \mathbb{R}$ and a variable indicator $v_j \in \{1, \ldots, V\}$, where $V$ represents the total number of variables. An IMTS sample can thus be represented as:

$$\mathcal{S}_i := \{(t_j, z_j, v_j)|j = 1, ..., M\}, \tag{1}$$

**Problem 1 (Irregular Multivariate Time Series Classification).** Given an IMTS sample $\mathcal{S}_i$, the problem is to correctly predict its class label $y_i$:

$$\mathcal{C}(\mathcal{S}_i) \to y_i, \tag{2}$$

where $\mathcal{C}(\cdot)$ denotes the classification model we aim to learn.

**Problem 2 (Irregular Multivariate Time Series Forecasting).** Given a split timestamp $t_S$, each sample $\mathcal{S}_i$ is segmented into a historical window $\mathcal{X}_i := \{(t_j, z_j, v_j)|j =$

$1, ..., M, t_j \leq t_S\}$ and a forecasting window $\mathcal{Y}_i := \{[(t_j, v_j), z_j]|j = 1, ..., M, t_j > t_S\}$. Elements $t_j$ and $v_j$ of $j$-th observation tuple in the set of forecasting windows are combined into a forecasting query $q_j \in \mathcal{Q}_i$. The problem is to accurately predict the corresponding observation values $\mathcal{Z}_i$ correspondence to forecasting query $\mathcal{Q}_i$ based on the historical window $\mathcal{X}_i$:

$$\mathcal{F}(\mathcal{X}_i, \mathcal{Q}_i) \to \mathcal{Z}_i, \tag{3}$$

where $\mathcal{F}(\cdot)$ denotes the forecasting model we aim to learn.

## 4. Methodology

In this section, we will take the $i$-th sample $S_i$ with $M_h$ historical observations as an example for introducing our method. We introduce the observation encoder in Section 4.1, followed by the introduction of the single intra-patch graph layer and inter-patch graph layer in Sections 4.2 and 4.3, respectively. Section 4.4 describes how the hierarchical architecture is implemented, and Section 4.5 covers the task decoder. Figure 2 presents the overall architecture of our model.

### 4.1. Observation Encoder

First, we introduce how to encode each observation into an $d_{\text{model}}$-dimensional graph node embedding. The $j$-th historical observation $o_j$ of $S_i$ corresponds to the tuple $(t_j, v_j, z_j)$, representing the observation value $z_j$ of variable $v_j$ at timestamp $t_j$. We encode these three parts separately.

For time encoding, we adopt continuous time embedding (Shukla & Marlin, 2021), which is designed specifically for IMTS. The $d$-th dimension of $d_{\text{model}}$-dimensional time embedding $\phi(t_j)$ for $t_j$ is calculated by:

$$\phi(t_j)[d] = \begin{cases} \omega_0 \cdot t + \alpha_0, & \text{if } d = 0 \\ \sin(\omega_d \cdot t + \alpha_d), & \text{if } 0 < d < d_{\text{model}} \end{cases}, \tag{4}$$

where $\omega_0, \alpha_0, \omega_d$ and $\alpha_d$ are learnable parameters. For variable encoding, we randomly initialize a learnable variable embedding for each variable, forming a variable embedding matrix $\boldsymbol{E} \in \mathbb{R}^{V \times d_{\text{model}}}$, the embedding corresponding to variable $v_j$ is $\boldsymbol{E}_{v_j}$. For value encoding, a linear layer $f(\cdot)$ is used to map the observation value $z_j$ into $d_{\text{model}}$-dimensional embedding.

With these three embeddings, the graph node embedding for $j$-th observation $o_j$ is calculated as:

$$h_j = \sigma[\phi(t_j) + \boldsymbol{E}_{v_j} + f(z_j)] \in \mathbb{R}^{d_{\text{model}}}, \tag{5}$$

where $\sigma(\cdot)$ is ReLU activation function.

### 4.2. Intra-Patch Graph Layer

In this section, we will introduce how to extract fine-grained features of densely sampled variables through an intra-patch

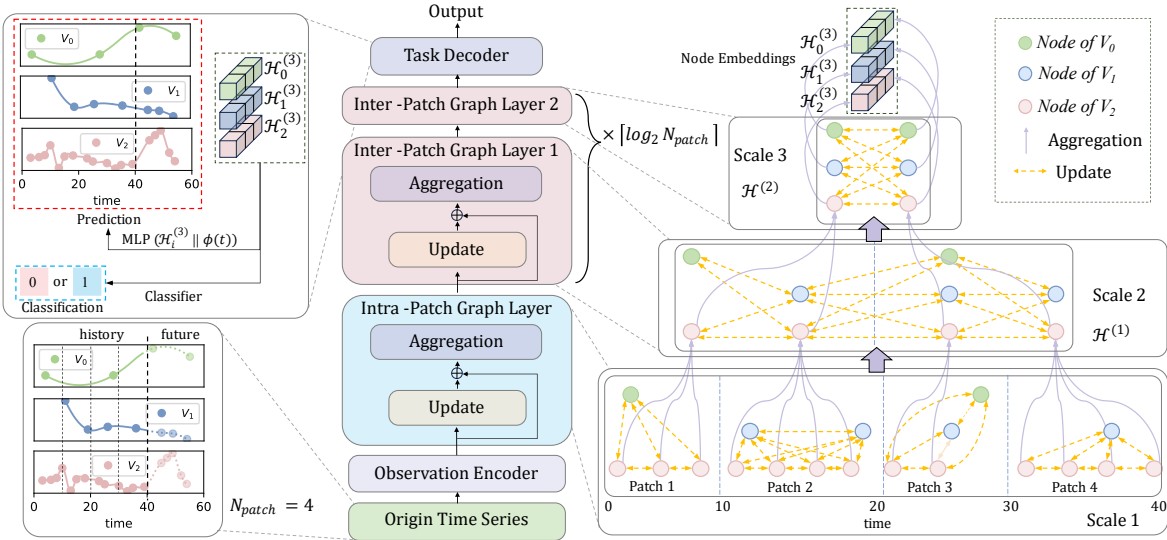

*Figure 2.* The model framework of Hi-Patch. The example input is an IMTS with three variables and a historical window of 0-40$s$ which is divided into $N = 4$ patches with a patch size $P = 10s$. First, each observation is encoded into a graph node through an observation encoder. Then, fine-grained features of densely sampled variables within each patch are extracted via an intra-patch graph layer and nodes are aggregated into patch-level nodes as inputs for multiple stacked inter-patch graph layers. After passing through $\lceil \log_2 N \rceil = 2$ inter-patch graph layers, coarse-grained features of all variables are progressively extracted and a single node embedding for each variable is obtained. Finally, the task decoder computes the downstream task output based on these node embeddings.

graph layer. First, we divide the historical observation nodes into several non-overlapping patches based on a short time span $P$. Given a total historical time span of $T$, it can be divided into $N = \lceil \frac{T}{P} \rceil$ patches. The $n$-th patch includes observation nodes of all variables within the period from $(n-1) \cdot P$ to $n \cdot P$, the index set of observation nodes in $n$-th patch is:

$$\mathcal{I}_n = \{j \mid j = 1, 2, ..., M_h, (n-1) \cdot P < t_j \leq n \cdot P\}. \quad (6)$$

The corresponding set of node states is $\mathcal{H}_n = \{h_j \mid j \in \mathcal{I}_n\}$, initialized by node embeddings. Each patch, divided based on a short time span, primarily consists of observation nodes from densely sampled variables, with a few or no observation nodes from sparsely sampled variables, thus forming a fine-grained view of densely sampled variables.

### 4.2.1. UPDATE

After obtaining several patches as fine-grained views of the densely sampled variables, each pair of nodes within a patch is connected by an edge to form a fully connected graph (intra-patch graph) for flexible representation of three types of local dependencies in IMTS in a unified manner, namely: 1) the same variable at different times, 2) different variables at the same time (synchronous) and 3) different variables at different times (asynchronous). The set of edges in $n$-th patch is:

$$\mathcal{E}_n = \{(u, v) \mid u \in \mathcal{I}_n, v \in \mathcal{I}_n\}. \quad (7)$$

We use graph attention network (GAT) (Veličković et al., 2018) to update the node states within each intra-patch graph for sufficiently extracting these dependencies. The formula for updating the state of node $v$ is:

$$h_v = h_v + \sum_{u \in \mathcal{N}(v)} \mathbf{MHA}(h_v, h_u, h_u), \quad (8)$$

where $\mathcal{N}(v)$ denotes the set of neighboring nodes of node $v$ and $\mathbf{MHA}$ denotes the multi-head attention mechanism (Vaswani et al., 2017):

$$\mathbf{MHA}(h_v, h_u, h_u) = Softmax\left(\frac{(W_k h_u)^T (W_q h_v)}{\sqrt{d_{\text{model}}}}\right) \cdot W_v h_u, \quad (9)$$

Due to the existence of three different types of dependencies, we adopted three sets of $\{W_q, W_k, W_v\}$ parameters to handle them separately. Compared to conventional methods that separately extract features along the time and variable dimensions, our intra-patch graph more comprehensively accounts for the asynchronous characteristics of IMTS. After the $L$-layer state update with Eq.(8), the local dependencies of the densely sampled variables at the original scale are thoroughly extracted.

### 4.2.2. AGGREGATION

To further extract features at a larger scale, we aggregate nodes of the same variables within each patch to create the

overall feature nodes of variables during the period of each patch. Specifically, for the nodes of the $v$-th variable in the $n$-th patch, the index set of nodes is $\mathcal{I}_{n,v} = \{j \mid j \in \mathcal{I}_n, v_j = v\}$. We first calculate the average observation timestamp as a reference time $\bar{t}_{n,v}^{(0)} = \frac{1}{|\mathcal{I}_{n,v}|} \sum_{j \in \mathcal{I}_{n,v}} t_j$, and then use multi-time attention (Shukla & Marlin, 2021) to aggregate the nodes to $\bar{t}_{n,v}^{(0)}$:

$$h_{n,v}^{(0)} = \sum_{j \in \mathcal{I}_{n,v}} \mathbf{MHA}(\phi(t_j), \phi(\bar{t}_{n,v}^{(0)}), h_j). \quad (10)$$

When a variable has no observation within a patch, no aggregation node is created, thus preserving the irregularities of the original series. We use a mask indicator $m_{n,v}^{(0)}$ to indicate whether the aggregation node exists of the $v$-th variable at the $n$-th patch. If $|\mathcal{I}_{n,v}| = 0$, $m_{n,v}^{(0)}$ is set as 0, else $m_{n,v}^{(0)} = 1$. The new set of node states for all $N$ patches of all $V$ variables after aggregation is $\mathcal{H}^{(0)} = \{h_{n,v}^{(0)} \mid n = 1, 2, ..., N, v = 1, 2, ..., V, m_{n,v}^{(0)} = 1\}$.

### 4.3. Inter-Patch Graph Layer

After passing through an intra-patch graph layer, fine-grained features with time scales less than $P$ of the original series have been extracted. In this section, we aim to extract features with time scale $P$ based on the aggregated node set $\mathcal{H}^{(0)}$ through an inter-patch graph layer.

#### 4.3.1. UPDATE

Each node in the set $\mathcal{H}^{(0)}$ represents the overall feature within time span $P$, and the reference time interval between nodes in adjacent patches is approximately equal to $P$. We connect the nodes located in adjacent patches pairwise in node set $\mathcal{H}^{(0)}$ by an edge to form a $P$-scale graph (inter-patch graph). This inter-patch graph flexibly represents the temporal and asynchronous inter-variable dependencies of densely sampled variables and some sparsely sampled variables with an origin sampling interval less than $P$ at scale $P$ in a unified manner. We continue to update the node states in this $P$-scale graph using GAT to extract the dependencies at scale $P$. The formula for updating is:

$$h_{n,v}^{(0)} = h_{n,v}^{(0)} + \sum_{n' \in \{n-1, n+1\}} \sum_{v'=1}^{V} \mathbf{MHA}(h_{n,v}^{(0)}, h_{n',v'}^{(0)}, h_{n',v'}^{(0)}). \quad (11)$$

#### 4.3.2. AGGREGATION

After the node state update, the features at scale $P$ are extracted. Subsequently, to extract features at scale $2P$, we aggregate every two nodes located on adjacent patches of the same variable by multi-time attention again to obtain the input of the next inter-patch graph layer. Specifically,

we calculate the average timestamp between two adjacent nodes as the reference time $\bar{t}_{n,v}^{(1)} = \frac{\bar{t}_{2n-1,v}^{(0)} + \bar{t}_{2n,v}^{(0)}}{m_{2n-1,v}^{(0)} + m_{2n,v}^{(0)}}$, where $n = 1, 2, ..., N/2$, $m_{2n-1,v}^{(0)} + m_{2n,v}^{(0)} > 0$, and then we aggregate the adjacent nodes to this reference time through multi-time attention to obtain feature nodes of $2P$ scale:

$$h_{n,v}^{(1)} = \sum_{j \in \{2n-1, 2n\}} \mathbf{MHA}(\phi(\bar{t}_{j,v}^{(0)}), \phi(\bar{t}_{n,v}^{(1)}), h_{j,v}^{(0)}). \quad (12)$$

The node state set after aggregating is $\mathcal{H}^{(1)} = \{h_{n,v}^{(1)} \mid n = 1, 2, ..., N/2, v = 1, 2, ..., V, m_{n,v}^{(1)} = 1\}$, which is served as input for the next inter-patch graph layer ($2P$-scale).

### 4.4. Hierarchical Architecture

Section 4.3 describes the first inter-patch graph layer ($P$-scale). This layer takes $\mathcal{H}^{(0)}$ as input, updates the node states, and aggregates them to $\mathcal{H}^{(1)}$ as the input of next inter-patch graph layer ($2P$ scale). Generalizing to the general case, the formula for the $l$-th inter-patch graph layer is:

$$\mathcal{H}^{(l)} = \text{Aggregation}(\text{Update}(\mathcal{H}^{(l-1)})), \quad (13)$$

where $\mathcal{H}^{(l)} = \{h_{n,v}^{(l)} \mid n = 1, 2, ..., N/2^l, v = 1, 2, ..., V, m_{n,v}^{(l)} = 1\}$ and $\mathcal{H}^{(l-1)} = \{h_{n,v}^{(l-1)} \mid n = 1, 2, ..., N/2^{l-1}, v = 1, 2, ..., V, m_{n,v}^{(l-1)} = 1\}$. The recursive form of Eq.(13) allows us to build a hierarchical architecture by stacking inter-patch graph layers, thereby further extracting features at scales of $2P, 4P, 8P, 16P, ....$ Until $l = \lceil \log_2 N \rceil$, the output of this layer is $\mathcal{H}^{(\lceil \log_2 N \rceil)} = \{h_{n,v}^{(\lceil \log_2 N \rceil)} \mid n = 1, v = 1, 2, ..., V, m_{n,v}^{(\lceil \log_2 N \rceil)} = 1\}$, where a single node embedding is obtained for each variable which has at least one observation. This process involves a total of $\lceil \log_2 N \rceil$ inter-patch graph layers, and fine-grained features of relatively densely sampled variables are extracted in the lower layers, while coarse-grained features of both densely and sparsely sampled variables are extracted in the upper layers. The node embedding of the $v$-th variable is $\mathcal{H}_v^{(\lceil \log_2 N \rceil)} \in \mathbb{R}^{d_{\text{model}}}$.

### 4.5. Task Decoder

#### 4.5.1. CLASSIFICATION

We calculate the sum of $d_{\text{model}}$ channels for each variable's node embedding to get a $V$-dimensional vector $C$ and use this to predict the final classification probabilities: $\hat{y} = \text{Softmax}(W^y C + b^y)$. The training objective is to minimize the cross-entropy loss between $\hat{y}$ and the ground truth $y$.

#### 4.5.2. FORECASTING

Given a query $q_j = (t_j, v_j)$, we follow (Zhang et al., 2024) by concatenating the node embedding of the query variable

$v_j$ with the embedding of the query time $t_j$, and then pass it through an MLP projection layer to generate the forecasting result: $\hat{z}_j = \text{MLP}([\mathcal{H}_{v_j}^{(\lceil \log_2 N \rceil)} \| \phi(t_j)])$. The training objective is to minimize the mean squared error loss between $\hat{z}_j$ and the ground truth $z_j$. The pseudo-code for Hi-Patch is presented in Appendix A (Algorithm 1).

# 5. Experiment

In this section, we present forecasting and classification experiments using a range of models and 8 datasets.

## 5.1. Experimental Setting

### 5.1.1. DATASETS AND BASELINES

For the forecasting task, we follow (Zhang et al., 2024) and use four datasets: PhysioNet (Silva et al., 2012), MIMIC-III (Johnson et al., 2016), Human Activity, and USHCN (Menne et al., 2015), covering the fields of healthcare, biomechanics, and climate science. We compare our method with seventeen relevant baselines, covering the SOTA models from (1) MTS forecasting: iTransformer (Liu et al., 2024), ModernTCN(Luo & Wang, 2024), TimesNet (Wu et al., 2023), PatchTST (Nie et al., 2023), (2) multi-scale MTS forecasting: Pathformer (Chen et al., 2024a) , TimeMixer (Wang et al., 2024b), MSGNet (Cai et al., 2024), MICN (Wang et al., 2023), (3) IMTS classification: Warpformer (Zhang et al., 2023), Raindrop (Zhang et al., 2022), GRU-D (Che et al., 2018), (4) IMTS forecasting: tPatchGNN (Zhang et al., 2024), GraFITi (Yalavarthi et al., 2024), CRU (Schirmer et al., 2022), mTAND (Shukla & Marlin, 2021), Neural Flows (Biloš et al., 2021), Latent ODEs (Rubanova et al., 2019).

For the classification task, we conduct experiments on four datasets in medical field where IMTS is most widely used, namely P19 (Reyna et al., 2020), PhysioNet (Silva et al., 2012), MIMIC-III (Johnson et al., 2016) and P12 (Goldberger et al., 2000) where PhysioNet is a reduced version of P12 considered by prior work (Shukla & Marlin, 2021). We compare our method with the state-of-the-art methods for irregular time series classification, including GRU-D (Che et al., 2018), ODE-RNN (Rubanova et al., 2019), IP-Net (Shukla & Marlin, 2019), SeFT (Horn et al., 2020), mTAND (Shukla & Marlin, 2021), Raindrop (Zhang et al., 2022), StraTS (Tipirneni & Reddy, 2022), DuETT (Labach et al., 2023), ViTST (Li et al., 2024a) and Warpformer (Zhang et al., 2023). In addition, we also compare our method with two approaches initially designed for forecasting tasks, namely DGM$^2$-O (Wu et al., 2021b) and MTGNN (Wu et al., 2020). The implementation and hyperparameter settings of these baselines are consistent with those of the original paper. More details of datasets and baselines can be found in Appendix B and C.

### 5.1.2. EVALUATION SETUP

For the forecasting task, we follow the data pre-processing method described in (Zhang et al., 2024) and randomly divide all the instances among each dataset into training, validation, and test sets according to ratios of 6:2:2. We use Mean Square Error (MSE) and Mean Absolute Error (MAE) to evaluate forecasting performance.

For the classification task, we follow the method described in (Harutyunyan et al., 2019) and divide the dataset into three parts for training, validation, and testing with the ratio of 70%,15%,15% on the MIMIC-III dataset. For the remaining three datasets, we adhered to (Zhang et al., 2022)'s approaches, and the ratio of training, validation, and testing set is 8:1:1. We measure the classification performance with the Area Under the Receiver Operating Characteristic Curve (AUROC) and Area Under the Precision-Recall Curve (AUPRC) since all the four datasets are binary classification datasets with highly imbalanced class distribution. More details of metrics can be found in Appendix D.

### 5.1.3. IMPLEMENTATION DETAILS

We adopt the Adam (Kingma & Ba, 2014) optimizer with a learning rate of 0.001, stopping it when the validation loss doesn't decrease over 10 epochs. All experiments are conducted with five random seeds, and the average and standard deviation are reported. All the models are experimented with using the PyTorch library on 2 GeForce RTX-3090-24G GPUs. The detailed settings of hyperparameters can be found in Appendix E.

## 5.2. Main Results

For the forecasting task, we test the model's performance under 3 varying observations and forecast horizons on each dataset. Table 1 shows the results of the default horizon, and the complete results are presented in Appendix F.1. Table 2 reports the models' classification performance on the other four datasets. In addition, to demonstrate the robustness of our method, we test whether Hi-Patch can achieve good classification performance when a subset of variables is completely missing. Table 9 (Appendix F.2) reports the results under different missing ratios. In summary, Hi-Patch achieves the best performance in 62 out of 72 metrics across 8 datasets for both classification and forecasting tasks.

Among these baselines, regular MTS forecasting models do not demonstrate competitive performance in IMTS forecasting since they have limited capability to handle the irregularities within and among variables in IMTS. In contrast, although our Hi-Patch is initially designed for IMTS, it also demonstrates competitive performance in regular MTS forecasting, as shown in Appendix F.3. As for the baselines specifically designed for IMTS, most of them focus on ex-

*Table 1.* Method benchmarking on IMTS forecasting. The best results are highlighted in **bold**, and the second-best results are in underlined. The results in the table are presented in the form of (Mean ± Std). '-' indicates a numerical overflow error.

| Methods | Human Activity (3000ms → 1000ms) | | USHCN (24months → 1month) | | PhysioNet (24h → 24h) | | MIMIC-III (24h → 24h) | |
|---|---|---|---|---|---|---|---|---|
| | $\text{MSE} \times 10^{-3}$ | $\text{MAE} \times 10^{-2}$ | $\text{MSE} \times 10^{-1}$ | $\text{MAE} \times 10^{-1}$ | $\text{MSE} \times 10^{-3}$ | $\text{MAE} \times 10^{-2}$ | $\text{MSE} \times 10^{-2}$ | $\text{MAE} \times 10^{-2}$ |
| iTransformer | 3.97 ± 0.10 | 4.30 ± 0.08 | 6.17 ± 0.07 | 4.18 ± 0.55 | 53.55 ± 19.59 | 16.87 ± 4.51 | 7.24 ± 0.50 | 21.24 ± 0.97 |
| ModernTCN | 3.99 ± 0.05 | 4.32 ± 0.04 | 5.83 ± 0.13 | 3.58 ± 0.09 | 28.99 ± 11.06 | 5.94 ± 0.20 | 10.07 ± 9.79 | 10.76 ± 1.10 |
| TimesNet | 3.79 ± 0.05 | 4.28 ± 0.04 | 5.62 ± 0.12 | 3.56 ± 0.12 | 9.30 ± 0.70 | 5.50 ± 0.34 | 2.34 ± 0.54 | 8.09 ± 0.10 |
| PatchTST | 5.21 ± 0.33 | 5.10 ± 0.20 | 5.88 ± 0.10 | 3.66 ± 0.13 | 25.56 ± 4.42 | 10.90 ± 1.08 | 7.24 ± 0.65 | 19.80 ± 0.87 |
| Pathformer | 3.40 ± 0.16 | 3.65 ± 0.08 | – | – | 6.75 ± 0.41 | 4.61 ± 0.20 | – | – |
| TimeMixer | 4.97 ± 0.31 | 5.02 ± 0.16 | 5.88 ± 0.10 | 3.59 ± 0.07 | 13.98 ± 0.31 | 6.88 ± 0.09 | 4.78 ± 0.09 | 14.29 ± 0.06 |
| MSGNet | 6.32 ± 0.16 | 6.06 ± 0.09 | 5.75 ± 0.06 | 3.64 ± 0.09 | 9.84 ± 0.29 | 5.79 ± 0.10 | 2.65 ± 0.18 | 9.10 ± 0.27 |
| MICN | 6.93 ± 0.12 | 6.04 ± 0.06 | 5.99 ± 0.11 | 3.69 ± 0.08 | 10.34 ± 0.24 | 6.00 ± 0.14 | 2.36 ± 0.06 | 8.43 ± 0.11 |
| Warpformer | 2.61 ± 0.02 | 3.12 ± 0.01 | 5.09 ± 0.03 | 3.10 ± 0.04 | 5.04 ± 0.14 | 3.72 ± 0.06 | 1.76 ± 0.30 | 7.27 ± 0.15 |
| Raindrop | 4.42 ± 0.25 | 4.65 ± 0.14 | 5.64 ± 0.10 | 3.29 ± 0.03 | 10.63 ± 0.29 | 6.02 ± 0.19 | 2.31 ± 0.07 | 8.61 ± 0.12 |
| GRU-D | 3.94 ± 0.29 | 4.37 ± 0.21 | 5.17 ± 0.06 | 3.21 ± 0.05 | 5.76 ± 0.34 | 4.53 ± 0.15 | 2.35 ± 0.06 | 8.34 ± 0.22 |
| tPatchGNN | 2.79 ± 0.09 | 3.24 ± 0.06 | 5.00 ± 0.03 | 3.07 ± 0.05 | 5.06 ± 0.10 | 3.75 ± 0.07 | 1.97 ± 0.05 | 7.76 ± 0.22 |
| GraFITi | 3.03 ± 0.14 | 3.45 ± 0.10 | 5.07 ± 0.03 | 2.97 ± 0.04 | 5.11 ± 0.19 | 3.96 ± 0.09 | 1.76 ± 0.04 | 7.28 ± 0.13 |
| CRU | 3.03 ± 0.04 | 3.60 ± 0.04 | 5.15 ± 0.50 | 3.18 ± 0.03 | 6.43 ± 0.62 | 4.51 ± 0.16 | 2.23 ± 0.03 | 7.99 ± 0.22 |
| mTAND | 3.14 ± 0.09 | 3.71 ± 0.06 | 5.03 ± 0.05 | 3.00 ± 0.06 | 6.18 ± 0.31 | 4.44 ± 0.19 | 2.15 ± 0.05 | 8.00 ± 0.06 |
| NeuralFlow | 4.29 ± 0.63 | 4.61 ± 0.43 | 5.41 ± 0.05 | 3.35 ± 0.06 | 7.68 ± 0.37 | 4.84 ± 0.19 | 2.34 ± 0.05 | 8.09 ± 0.09 |
| Latent-ODE | 3.32 ± 0.10 | 3.91 ± 0.08 | 5.16 ± 0.04 | 3.21 ± 0.07 | 6.85 ± 0.28 | 4.77 ± 0.17 | 2.11 ± 0.15 | 7.76 ± 0.08 |
| **Hi-Patch** | **2.57 ± 0.02** | **3.11 ± 0.03** | **4.94 ± 0.05** | **2.96 ± 0.04** | **4.86 ± 0.03** | **3.62 ± 0.07** | **1.75 ± 0.26** | **7.24 ± 0.18** |

*Table 2.* Method benchmarking on IMTS classification. The best results are highlighted in **bold**, and the second-best results are in underlined. The results in the table are presented in the form of (Mean ± Std %).

| Methods | P19 | | PhysioNet | | MIMIC-III | | P12 | |
|---|---|---|---|---|---|---|---|---|
| | AUROC | AUPRC | AUROC | AUPRC | AUROC | AUPRC | AUROC | AUPRC |
| GRU-D | 88.7 ± 1.2 | 57.6 ± 2.3 | 79.1 ± 6.9 | 42.7 ± 7.2 | 82.2 ± 1.8 | 43.3 ± 2.1 | 79.6 ± 0.6 | 41.7 ± 1.8 |
| ODE-RNN | 87.1 ± 1.0 | 52.6 ± 3.2 | 75.5 ± 2.8 | 33.7 ± 4.1 | 81.0 ± 0.6 | 42.3 ± 0.7 | 78.8 ± 0.6 | 37.4 ± 2.6 |
| IP-Net | 90.2 ± 0.2 | 58.6 ± 0.8 | 86.8 ± 0.6 | 55.8 ± 1.4 | 84.1 ± 0.1 | 47.1 ± 0.9 | 83.7 ± 0.3 | 46.3 ± 1.3 |
| SeFT | 84.0 ± 0.3 | 49.3 ± 0.5 | 75.5 ± 0.2 | 29.4 ± 0.9 | 67.9 ± 0.2 | 23.2 ± 0.4 | 78.1 ± 0.5 | 35.9 ± 0.8 |
| MTGNN | 88.5 ± 1.0 | 55.8 ± 1.5 | 77.1 ± 4.4 | 35.4 ± 7.3 | 78.5 ± 2.3 | 35.2 ± 3.1 | 82.1 ± 1.5 | 41.8 ± 2.1 |
| mTAND | 82.9 ± 0.9 | 32.2 ± 1.5 | 86.8 ± 1.3 | 52.5 ± 1.3 | 83.8 ± 0.3 | 46.6 ± 0.5 | 85.3 ± 0.3 | 49.3 ± 1.0 |
| DGM$^2$-O | 91.6 ± 0.5 | 60.0 ± 1.3 | 85.8 ± 0.7 | 50.4 ± 3.2 | 80.4 ± 0.5 | 36.0 ± 0.8 | 85.8 ± 0.1 | 48.3 ± 0.7 |
| Raindrop | 87.6 ± 2.7 | 61.1 ± 1.4 | 81.2 ± 0.9 | 37.3 ± 2.0 | 79.8 ± 1.3 | 35.2 ± 1.1 | 82.0 ± 0.6 | 42.7 ± 1.7 |
| StraTS | 91.2 ± 0.3 | 58.4 ± 1.4 | 84.9 ± 1.5 | 47.3 ± 5.3 | 84.4 ± 0.4 | 46.4 ± 0.8 | 86.7 ± 0.7 | 52.1 ± 1.5 |
| DuETT | 88.2 ± 0.5 | 56.0 ± 3.9 | 81.3 ± 1.4 | 44.9 ± 1.4 | 78.8 ± 0.8 | 34.3 ± 1.0 | 83.4 ± 1.2 | 45.4 ± 1.5 |
| ViTST | 91.7 ± 0.1 | 57.5 ± 0.7 | 81.3 ± 1.9 | 37.4 ± 2.9 | 81.8 ± 0.3 | 39.6 ± 1.3 | 86.3 ± 0.1 | 50.8 ± 1.5 |
| Warpformer | 91.8 ± 0.4 | 60.6 ± 2.6 | 83.3 ± 0.7 | 43.5 ± 2.3 | 84.6 ± 0.3 | 46.6 ± 0.9 | 85.4 ± 0.5 | 50.4 ± 1.5 |
| **Hi-Patch** | **92.1 ± 0.4** | **61.1 ± 2.1** | **86.8 ± 0.9** | **57.3 ± 1.9** | **84.8 ± 0.2** | **47.2 ± 1.0** | **86.9 ± 0.7** | **53.3 ± 0.9** |

tracting correlations at the origin scale, failing to achieve comprehensive multi-scale feature extraction and resulting in suboptimal performance. Additionally, we conduct experiments with a fixed prediction horizon while varying the observation horizons to demonstrate our model's ability to learn more temporal dependencies from long-term data in Appendix F.4. Furthermore, we provide the computational efficiency analysis in Appendix G and comprehensive discussions on the technical details of Hi-Patch and existing methods in Appendix H.

### 5.3. Ablation Study

In this section, we investigate the performance benefits generated by each key component of the proposed method on the forecasting task. We compare the Hi-Patch with its four variants: (1) **w/o Hie**: We remove the hierarchical multi-scale architecture and set the patch size to the time span

*Table 3.* Ablation results of Hi-Patch on two datasets. The results in the table are presented in the form of (Mean ± Std).

| Methods | Hunam Activity | | USHCN | |
|---|---|---|---|---|
| | $\text{MSE} \times 10^{-3}$ | $\text{MAE} \times 10^{-2}$ | $\text{MSE} \times 10^{-1}$ | $\text{MAE} \times 10^{-1}$ |
| **Hi-Patch** | **2.57 ± 0.02** | **3.11 ± 0.03** | **4.94 ± 0.05** | **2.96 ± 0.04** |
| w/o Hie | 2.70 ± 0.04 | 3.13 ± 0.03 | 5.35 ± 0.13 | 3.26 ± 0.13 |
| w/o DVDT | 2.68 ± 0.04 | 3.12 ± 0.01 | 5.21 ± 0.03 | 3.11 ± 0.12 |
| w/o 3W | 2.60 ± 0.01 | 3.17 ± 0.03 | 4.99 ± 0.03 | 3.08 ± 0.10 |
| w/o TEAGG | 2.77 ± 0.02 | 3.22 ± 0.03 | 5.23 ± 0.04 | 3.13 ± 0.03 |

of the entire historical window, only extracting features at the original scale. (2) **w/o DVDT**: We removed the asynchronous edges of different variables at different times in intra/inter patch graph layers, retaining only the edges between nodes of the same variable at different times and different variables at the same time; (3) **w/o 3W**: We use only one set of attention parameter matrices $\{W_q, W_k, W_v\}$ for three types of edges; (4) **w/o TEAGG**: We aggregate

nodes with the same variables within a patch using mean aggregation rather than multi-time attention aggregation. The results on the Human Activity dataset and USHCN dataset are presented in Table 3, while the results for the remaining datasets are presented in Table 11 (Appendix I). The results show that all components of Hi-Patch are necessary.

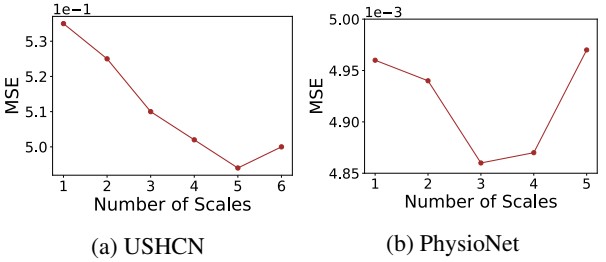

(a) USHCN

(b) PhysioNet

*Figure 3.* Effect of different scale quantities on two datasets.

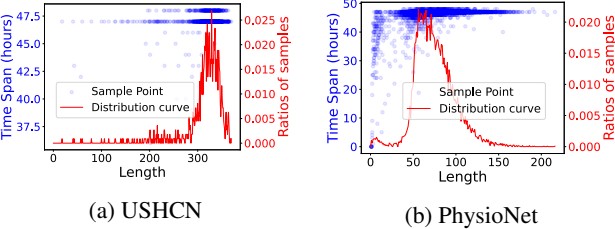

(a) USHCN

(b) PhysioNet

*Figure 4.* Distribution of sample length and time span on two datasets. Each blue dot represents a sample, with its x-coordinate indicating the sample length and the left y-axis representing its time span. The red curve is the distribution curve of sample length.

### 5.4. Effect of Scale Quantity

Figures 3 and 8 illustrate the impact of different scale quantities on four datasets. Generally, increasing the number of scales improves performance within a certain range. However, beyond a certain point, further increasing the number of scales has a negative effect. This is because the number of scales is inversely related to patch size in our method: as the number of scales increases, the patch size decreases. A patch that is too small may not contain enough observations to extract local patterns effectively. Additionally, we observe that on the PhysioNet and MIMIC-III datasets, performance with too many scales is even worse than with a single scale. To further investigate, we visualize the distribution of sample lengths and time spans across the four datasets in Figures 4 and 9. We find that samples in PhysioNet and MIMIC-III primarily exhibit a 'short length, long span' characteristic, which indicates that samples in these two datasets predominantly show coarse-grained patterns with few fine-grained local features. In such cases, using too many scales becomes redundant and can significantly degrade performance. We provide visualizations of the multi-scale views at different layers in our model in Appendix K.

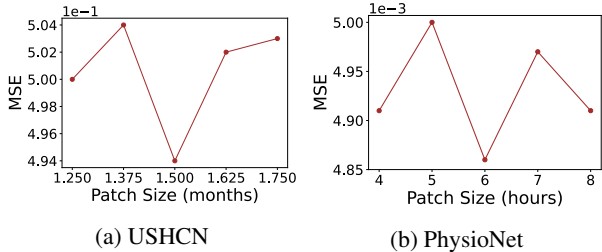

(a) USHCN

(b) PhysioNet

*Figure 5.* Effect of different patch size

### 5.5. Effect of Patch Size

The impact of patch size on the performance of our model in two real-world datasets, USHCN (climate) and PhysioNet (clinical), is illustrated in Figure 5. For the USHCN dataset, the optimal model performance is achieved at a patch size of 1.5 months. At this patch size, our hierarchical architecture extracts features at scales of 1.5 months, 3 months, 6 months, and 12 months, which precisely cover several important observational scales in climatology, including the seasonal (Doblas-Reyes et al., 2013) (3 months), monsoonal (Clift & Plumb, 2008) (6 months) and annual (Almazroui et al., 2012) cycle (12 months). For the PhysioNet dataset, the optimal patch size is 6 hours. At this patch size, our model extracts features at the 6-hour, 12-hour, and 24-hour scales, which also align with actual cycles in clinical medicine. Among them, 6 hours is a common clinical monitoring period used in medical practice (Seymour et al., 2017), while 12-hour and 24-hour cycles reflect circadian rhythms and daily cycles, which are crucial for assessing patients' physiological changes and disease fluctuations (Klerman et al., 2022). This alignment with real-world periodicity enables it to better capture the inherent temporal dynamics in diverse applications. By aligning with critical observational and diagnostic timeframes, our model enhances its predictive power and interpretability, making it highly adaptable and effective across practical scenarios that require nuanced temporal understanding.

## 6. Limitations

Although our proposed method introduces a multi-scale framework for sparse IMTS, it exhibits certain limitations in terms of scalability. As detailed in Appendix G, the primary computational cost of our approach stems from the intra-patch graph layer, whose complexity scales quadratically with the number of observed points within each patch. While this cost is acceptable for sparsely sampled IMTS data, it becomes prohibitive when applied to large-scale datasets with dense observations. Consequently, a promising direction for future work is to reduce the computational complexity of the intra-patch graph layer in Hi-Patch to improve its scalability. Appendix G.3 provides further dis-

cussion on this topic.

## 7. Conclusion

In this paper, we propose Hi-Patch for modeling IMTS. The proposed method leverages the intra-patch/inter-patch graph neural network to flexibly represent and fully extract features at specific scales in IMTS. Based on this, the hierarchical architecture is used to effectively achieve multi-scale modeling of IMTS in a bottom-up manner (from local to global). Experimental results demonstrate that Hi-Patch outperforms existing methods in both IMTS forecasting and classification tasks. Our future work will focus on adaptive multi-scale modeling of IMTS, which selects the most suitable scales based on the specific temporal characteristics and dynamics of each sample to further improve performance.

## Impact Statement

This paper presents work whose goal is to advance the field of Machine Learning. There are many potential societal consequences of our work, none which we feel must be specifically highlighted here.

## Acknowledgements

The work described in this paper was partially funded by the National Natural Science Foundation of China (Grant No. 62272173), the Natural Science Foundation of Guangdong Province (Grant Nos. 2024A1515010089, 2022A1515010179), the Science and Technology Planning Project of Guangdong Province (Grant No. 2023A0505050106), and the National Key R&D Program of China (Grant No. 2023YFA1011601).

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

# A. Algorithm of Hi-Patch

---

**Algorithm 1** The pseudo-code of Hi-Patch for forecasting

---

**Input**: An IMTS sample $\mathcal{S}_i$ with $M$ observations, a split time $t_S$ historical window $\mathcal{X}_i := \{(t_j, z_j, v_j)|j = 1, ..., M, t_j \leq t_S\}$, forecasting query $\mathcal{Q}_i := \{[(t_j, v_j)]|j = 1, ..., M, t_j > t_S\}$, patch size $P$, total time span $T$
**Output**: Predicted value set $\hat{\mathcal{Z}}_i$

1: ▷ Observation Encoder
2: **for** $j = 1, 2, ..., |\mathcal{X}_i|$ **do**
3:    encode observation tuple $o_j = (t_j, z_j, v_j)$ as graph node embedding $h_j$ using Eq.(5)
4: **end for**
5: ▷ Intra-Patch Graph Layer
6: **for** $p = 1, 2, ..., \lceil \frac{T}{P} \rceil$ **do**
7:    construct intra-patch graph in patch $p$ using Eq.(6) and (7)
8:    update states of intra-patch graph nodes in patch $p$ through GAT using Eq.(8) and Eq.(9)
9:    aggregate nodes of the same variable in patch $p$ through multi-time attention using Eq.(10), get $\mathcal{H}^{(0)}$
10: **end for**
11: ▷ Inter-Patch Graph Layers
12: **for** $l = 1, 2, ..., \lceil \log_2 \lceil \frac{T}{P} \rceil \rceil$ **do**
13:    construct $lP$-scale inter-patch graph
14:    update states of $lP$-scale inter-patch graph nodes through GAT using Eq.(11)
15:    aggregate nodes of every two adjacent nodes of the same variable in $lP$-scale inter-patch graph through multi-time attention using Eq.(12), get $\mathcal{H}^{(l)}$
16: **end for**
17: ▷ Task Decoder
18: **for** $j = 1, 2, ..., |\mathcal{Q}_i|$ **do**
19:    $\hat{z}_j = \text{MLP}([\mathcal{H}_{v_j}^{(\lceil \log_2 \lceil \frac{T}{P} \rceil \rceil)} \parallel \phi(t_j)])$
20:    $\hat{\mathcal{Z}}_i \leftarrow \hat{\mathcal{Z}}_i \cup \{\hat{z}_j\}$
21: **end for**
22: **return** $\hat{\mathcal{Z}}_i$

---

# B. Datasets

We use 8 irregularly multivariate time series datasets to evaluate the performance of our model and baseline models. The dataset statistics are summarized in Table 4.

*Table 4.* Dataset statistics.

| Task | Datasets | # Samples | # Variables | Avg Sample Length | Missing Ratio |
|---|---|---|---|---|---|
| | Human Activity | 5400 | 12 | 120 | 75% |
| Forecasting | USHCN | 26736 | 5 | 163 | 77.9% |
| | PhysioNet | 11988 | 36 | 74 | 88.4% |
| | MIMIC-III | 23457 | 96 | 46 | 96.7% |
| | P19 | 38803 | 34 | 36 | 94.9% |
| Classification | P12 | 11988 | 36 | 74 | 88.4% |
| | MIMIC-III | 21107 | 16 | 78 | 65.5% |
| | PhysioNet | 3997 | 36 | 74 | 84.9% |

## B.1. Forecasting

For the forecasting task, we use four datasets and follow (Zhang et al., 2024)'s data preprocessing program. Here is the detailed information of these datasets. We use three historical/forecasting horizons on each dataset.

**PhysioNet**    (Silva et al., 2012) This dataset includes 12000 IMTS from different patients, each with 41 clinical signals collected irregularly during the first 48 hours of ICU admission. We use the first 24/36/12 hours as the observed data to predict the queried values in the subsequent 24/12/36 hours.

**MIMIC-III**    (Johnson et al., 2016) MIMIC-III is a clinical database containing IMTS data from 23457 patients, each with 96 variables recorded during the first 48 hours of ICU admission. We use the first 24/36/12 hours as the observed data to predict the queried values in the subsequent 24/12/36 hours.

**Human Activity**    This dataset consists of 12 irregularly measured 3D positional variables from sensors worn on the ankles, belts, and chests of five individuals performing various activities. To better align with realistic forecasting scenarios, the original time series is chunked into 5400 IMTS, each spanning 4000 milliseconds. The first 3000/2000/1000 milliseconds are used as observed data to predict the sensor positions for the next 1000/2000/3000 milliseconds.

**USHCN**    (Menne et al., 2015) The USHCN dataset includes over 150 years of climate data from multiple U.S. stations, covering 5 climate variables. Following established preprocessing methods, we focus on data from 1114 stations between 1996 and 2000, resulting in 26736 IMTS. Each instance uses data from the previous 24 months to predict the next 1/6/12 month's climate conditions.

### B.2. Classification

**P19**    (Reyna et al., 2020) The PhysioNet Sepsis Early Prediction Challenge 2019 dataset contains medical records of 38,803 patients. Each record includes 34 variables and a static vector detailing attributes such as age, gender, time between hospital and ICU admission, ICU type, and ICU length of stay in days. Each patient also receives a binary label indicating whether sepsis occurs within the next 6 hours. We exclude samples with excessively short or long time series following (Zhang et al., 2022). Available at https://physionet.org/content/challenge-2019/1.0.0/.

**P12**    (Goldberger et al., 2000) The P12 dataset comprises data from 11,988 patients after removing 12 inappropriate samples identified by (Horn et al., 2020). Each record includes multivariate time series data from the first 48 hours of ICU stay, consisting of 36 sensor measurements (excluding weight) and a static vector with 9 elements, including age and gender. Patients are labeled based on ICU stay duration: a negative label indicates three days or less, and a positive label indicates more than three days. Available at https://physionet.org/content/challenge-2012/1.0.0/.

**MIMIC-III**    (Johnson et al., 2016) MIMIC-III is a widely used dataset containing de-identified EHRs of ICU patients admitted to Beth Israel Deaconess Medical Center from 2001 to 2012, originally with around 57,000 records covering variables such as medications and vital signs. We focus on the in-hospital mortality prediction task, using a subset established by (Harutyunyan et al., 2019). After preprocessing, our dataset includes 16 features and 21,107 data points. Available at https://physionet.org/content/mimiciii/1.4/.

**PhysioNet**    (Silva et al., 2012) Physionet contains the data from the first 48 hours of ICU patients, which is a reduced version of P12 considered by prior work. Therefore, we follow the same preprocessing methods used for the P12 dataset. The processed data set includes 3997 labeled instances. We focus on predicting in-hospital mortality.

## C. Baselines

### C.1. Forecasting

#### C.1.1. METHODS FOR REGULAR MTS

For methods for regular MTS, we organize each sample into a $V \times T_h$ history matrix and a $V \times T_f$ forecasting matrix, where $V$ represents the maximum number of variables and $T_h/T_f$ represents the historical/forecasting length of the sample. Unobserved positions are filled with zeros, the sequence length is set to the maximum historical length across all samples, and the forecasting length is set to the maximum forecasting length across all samples.

**iTransformer**    (Liu et al., 2024) uses inverted Transformers for time series forecasting. We use the following setting in our experiment: $e_{\text{layers}} = 3$, $d_{\text{layers}} = 1$, factor $= 3$, $d_{\text{model}} = 512$, $d_{\text{ff}} = 512$.

**TimesNet** (Wu et al., 2023) analyses general time series through temporal 2D-Variation. We use the following setting in our experiment: $e_{\text{layers}} = 2$, $d_{\text{layers}} = 1$, factor $= 3$, $d_{\text{model}} = 256$, $d_{\text{ff}} = 512$, top-k $= 5$.

**PatchTST** (Nie et al., 2023) is a Transformer-based model using patch and channel independence for long-term time series forecasting. We use the following setting in our experiment: $e_{\text{layers}} = 2$, $d_{\text{layers}} = 1$, factor $= 3$, patch_len $= 16$, stride $= 8$.

**MICN** (Wang et al., 2023) achieves multi-scale local and global context modeling for long-term time series forecasting. We use the following setting in our experiment: $e_{\text{layers}} = 2$, $d_{\text{layers}} = 1$, factor $= 3$, $d_{\text{model}} = 256$, $d_{\text{ff}} = 512$, top_k $= 5$, decomp_kernel $= 32$, conv_kernel $= 24$, isometric_kernel $= [18, 6]$.

**TimeMixer** (Wang et al., 2024b) achieves complementary predictive capabilities by disentangling variations in multi-scale series. We use the following setting in our experiment: $e_{\text{layers}} = 3$, $d_{\text{layers}} = 1$, factor $= 3$, $d_{\text{model}} = 16$, $d_{\text{ff}} = 32$, down_sampling_layers $= 3$, down_sampling_window $= 2$.

For the above 5 methods, we use the implementation in Time-Series-Library

**ModernTCN** (Luo & Wang, 2024) is a modern pure convolution structure designed for general time series analysis. We use the implementation in https://github.com/luodhhh/ModernTCN and the following setting in our experiment: ffn_ratio $= 8$, patch_size $= 8$, patch_stride $= 4$, num_blocks $= 1$, large_size $= 51$, small_size $= 5$, dims $= 64$

**Pathformer** (Chen et al., 2024a) is a multi-scale transformer architecture with adaptive pathways for time series forecasting. We use the implementation in https://github.com/decisionintelligence/pathformer and the following setting in our experiment: $k = 2$, layer_nums $= 3$, $d_{\text{model}} = 16$, $d_{\text{ff}} = 64$.

**MSGNet** (Cai et al., 2024) learns multi-scale inter-series correlations for multivariate time series forecasting. We use the implementation in https://github.com/YoZhibo/MSGNet and the following setting in our experiment: $e_{\text{layers}} = 2$, $d_{\text{model}} = 512$, $d_{\text{ff}} = 64$, n_heads $= 8$, top_k $= 5$, dropout $= 0.1$, nums_kernels $= 6$, conv_channel $= 32$, skip_channel $= 32$, gcn_depth $= 2$, propalpha $= 0.3$, node_dim $= 10$, gcn_dropout $= 10$.

### C.1.2. METHODS FOR IMTS

IMTS forecasting methods can directly make predictions. For IMTS classification methods, we use them as encoders to extract variable-level representations for each sample, followed by forecasting using the decoder described in Section 4.5.2.

**Warpformer** (Zhang et al., 2023) A transformer-based network that captures features at different scales in IMTS using warping modules and dual attention mechanisms. We use three scales with normalized length $\widetilde{L}^{(0)} = 0$, $\widetilde{L}^{(1)} = 0.2$ and $\widetilde{L}^{(2)} = 1$. The dimension of representations $D$ is set as $32$. The attention heads and the layers of the warpformer are set as 1 and 2, respectively. We use the implementation at https://github.com/imJiawen/Warpformer.

**Raindrop** (Zhang et al., 2022) A graph neural network that embeds IMTS while learning the dynamics of sensors purely from observation data. We use the following setting in our experiment: $d_{ob} = 4$, $p^t = 16$, $r_v = 16$, $L = 2$, $d_k = 20$, $d_a = V$. We use the implementation at https://github.com/mims-harvard/Raindrop.

**GRU-D** (Che et al., 2018) GRU-D takes two representations of missing patterns, i.e., masking and time interval, and effectively incorporates them into a deep model architecture. The number of hidden states of GRU-D is set as 49. We use the implementation from https://github.com/Han-JD/GRU-D.

**tPatchGNN** (Zhang et al., 2024) is a transformable patching graph neural network for IMTS forecasting. We use the implementation in https://github.com/usail-hkust/t-PatchGNN and use the hyperparameters specified in their scripts.

**GraFITi** (Yalavarthi et al., 2024) use bipartite graph for representing and forecasting of IMTS. We use the following setting in our experiment: latent_dim $= 128$, n_layers $= 4$, $n_{\text{heads}} = 1$. We use the implementation at https://github.com/yalavarthivk/GraFITi.

**CRU** (Schirmer et al., 2022) models IMTS with continuous recurrent units. We use the following setting in our experiment: latent_state_dim = 20, hidden_units = 50, bandwidth = 10, num_basis = 20, trans_covar = 0.1. We use the implementation at https://github.com/boschresearch/Continuous-Recurrent-Units.

**mTAND** (Shukla & Marlin, 2021) A deep learning framework for IMTS data that learns an embedding of continuous time values and uses an attention mechanism to produce a fixed-length representation. We set the latent dimension and the hidden size of GRU to 32. The number of reference points and the dimension of time embedding is 128. We use the implementation at https://github.com/reml-lab/mTAN.

**NeuralFlows** (Biloš et al., 2021) use neural networks to model ODE solution curves to mitigate the expensive numerical solvers in neural ODEs. We use the implementation at https://github.com/mbilos/neural-flows-experiments and the following setting in our experiment: flow_model='CouplingFlow', hidden_dim = 50, hidden_layers = 3, latents = 20, rec_dims = 40.

**Latent-ODEs** (Rubanova et al., 2019) is an ODE-based model that improves RNNs with continuous-time hidden state dynamics specified by neural ODEs. We use the implementation in https://github.com/YuliaRubanova/latent_ode and the following setting in our experiment: latents = 20, units = 50, gen_layers = 3, rec_dims = 40, rec_layers = 3, gru_units = 50.

### C.2. Classification

**ODE-RNN** (Rubanova et al., 2019) ODE-RNN uses neural ODEs to model hidden state dynamics and an RNN to update the hidden state in the presence of a new observation. The latent dimension is set as 40, and the ODE function has 3 layers with 50 units. We use the implementation at https://github.com/YuliaRubanova/latent_ode

**SEFT** (Horn et al., 2020) A set function approach where all the observations are modeled individually before pooling them together using an attention-based approach. We use a constant architecture for the attention network $f'$ with 2 layers, 4 heads and dimensionality of the dot product space $d$ of 128. In addition, the attention network $f'$ was always set to use mean aggregation. We use the implementation from https://github.com/BorgwardtLab/SeFT.

**IP-Net** (Shukla & Marlin, 2019) A model architecture for IMTS data based on several semi-parametric interpolation layers organized into an interpolation network followed by a prediction network GRU. The number of reference points is set as 192. The hidden size of GRU is 100. We take the source code at https://github.com/mlds-lab/interp-net.

**DGM$^2$-O** (Wu et al., 2021b) A generative model, which tracks the transition of latent clusters instead of isolated feature representations, achieves robust sparse time series modeling. We use the DGM$^2$-O and set both the hidden dimension and the cluster_num as 10. We use the source code at https://github.com/thuwuyinjun/DGM2.

**MTGNN** (Wu et al., 2020) A general graph neural network framework designed for MTS data. We use 5 graph convolution and 5 temporal convolution modules with the dilation exponential factor 2. The graph convolution and temporal convolution modules have 16 output channels. The skip connection layers all have 32 output channels. The first layer of the output module has 64 output channels, and the second layer has 1 output channel. We use the implementation from: https://github.com/nnzhan/MTGNN.

**StraTS** (Tipirneni & Reddy, 2022) is a self-supervised transformer for sparse IMTS. We use the implementation at https://github.com/sindhura97/STraTS and the following setting in our experiment: hidden_dim = 64, num_layers = 2, num_heads = 16, dropout = 0.2.

**DuETT** (Labach et al., 2023) is a dual event time transformer for Electronic Health Records (EHRs). We use the implementation at https://github.com/layer6ai-labs/DuETT and the default settings of the model declaration in this repository.

**ViTST** (Li et al., 2024a): transforms IMTS into line graph images and adapts powerful vision transformers to perform time series classification in the same way as image classification. We use the implementation at https://github.com/Leezekun/ViTST.

Please refer to C.1.2 for details of **GRU-D** (Che et al., 2018), **mTAND** (Shukla & Marlin, 2021) , **Raindrop** (Zhang et al., 2022) and **Warpformer** (Zhang et al., 2023).

## D. Performance Metrics

**MSE**    MSE (Mean Squared Error) measures the average of the squared differences between predicted and actual values. The calculation formula is:

$$\text{MSE} = \frac{1}{n} \sum_{i=1}^{n} (y_i - \hat{y}_i)^2, \tag{14}$$

where $y_i$ represents the actual value, $\hat{y}_i$ represents the predicted value, and $n$ is the number of observations. A smaller MSE indicates better model performance. Since errors are squared, MSE is sensitive to large errors or outliers.

**MAE**    MAE (Mean Absolute Error) measures the average of the absolute differences between predicted and actual values. The calculation formula is:

$$\text{MAE} = \frac{1}{n} \sum_{i=1}^{n} |y_i - \hat{y}_i|, \tag{15}$$

where $y_i$ represents the actual value, $\hat{y}_i$ represents the predicted value, and $n$ is the number of observations. A smaller MAE indicates better model performance. Compared to MSE, MAE is less sensitive to outliers and provides a straightforward average measure of error.

**AUROC**    AUROC is commonly employed in binary classification tasks, where one class is designated as positive and the other as negative. It represents the area under the Receiver Operating Characteristic (ROC) curve, constructed by plotting the True Positive Rate (TPR) against the False Positive Rate (FPR). AUROC ranges from 0 to 1, with a higher value indicating better model performance in accurately discriminating between positive and negative instances. An AUROC equal to 0.5 indicates a model's performance equivalent to random guessing, while an AUROC greater than 0.5 signifies superiority over random guessing.

**AUPRC**    The Area Under the Precision-Recall Curve is widely used as a performance metric for imbalanced binary classification tasks. It provides a comprehensive assessment of a model's precision-recall trade-off. The Precision-Recall curve is constructed by plotting recall on the x-axis and precision on the y-axis. AUPRC ranges from 0 to 1, and a higher value indicates better model performance in achieving high precision and recall simultaneously. It has been suggested as a good criterion for unevenly distributed classification problems (Davis & Goadrich, 2006).

## E. Hyperparameters Settings

We search all hyperparameters in the grid for our proposed model Hi-Patch. Specifically, our model has a total of 4 hyperparameters: patch size $P$, dimension of node state $d_{\text{model}}$, number of multi-head attention heads $n_{\text{heads}}$, number of GAT layers $L$. Since the number of patches $N = \lceil \frac{T}{P} \rceil$ and the number of inter-patch graph layers equals to $\lceil \log_2 N \rceil$ where $T$ is the dataset-specific maximum time span, we search $N$ over the range $\{2, 4, 8, 16, 32\}$ to maintain the number of inter-patch graph layers as an integer. Thus the patch size $P$ of each dataset is determined by $P = \lceil \frac{T}{N} \rceil$. Additionally, we search $d_{\text{model}}$ in $\{16, 32, 64, 128\}$, $n_{\text{heads}}$ in $\{1, 2, 4, 8\}$ and $L$ in $\{1, 2, 3\}$. The best hyperparameters for each dataset are reported in the code.

# F. Additional Experiments

## F.1. Varying Observation and Forecast Horizons

*Table 5.* Performance of varying observation and forecast horizons on Human Activity dataset. The best results are highlighted in **bold**, and the second-best results are in underlined. The results in the table are presented in the form of (Mean ± Std).

| Horizon | 2000ms → 2000ms | | 1000ms → 3000ms | |
|---|---|---|---|---|
| Metric | MSE$\times 10^{-3}$ | MAE$\times 10^{-2}$ | MSE$\times 10^{-3}$ | MAE$\times 10^{-2}$ |
| iTransformer | 7.49 ± 4.72 | 6.08 ± 2.17 | 5.58 ± 0.04 | 5.13 ± 0.05 |
| ModernTCN | 5.26 ± 0.06 | 4.96 ± 0.04 | 12.12 ± 1.01 | 5.72 ± 0.07 |
| TimesNet | 5.38 ± 0.30 | 5.32 ± 0.20 | 9.90 ± 0.42 | 7.34 ± 1.81 |
| PatchTST | 7.25 ± 0.29 | 6.26 ± 0.17 | 8.97 ± 1.96 | 6.94 ± 0.94 |
| Pathformer | 4.67 ± 0.22 | 4.58 ± 0.17 | 5.49 ± 0.12 | 5.05 ± 0.06 |
| TimeMixer | 5.39 ± 0.54 | 5.05 ± 0.38 | 5.96 ± 0.19 | 5.36 ± 0.10 |
| MSGNet | 7.90 ± 0.36 | 6.85 ± 0.13 | 8.29 ± 0.22 | 7.02 ± 0.12 |
| MICN | 7.57 ± 0.05 | 6.43 ± 0.02 | 8.16 ± 0.12 | 6.78 ± 0.05 |
| Warpformer | 3.60 ± 0.08 | 3.81 ± 0.03 | 4.26 ± 0.11 | 4.26 ± 0.04 |
| Raindrop | 5.57 ± 0.34 | 5.15 ± 0.11 | 5.75 ± 0.33 | 5.37 ± 0.22 |
| GRU-D | 5.93 ± 0.10 | 5.66 ± 0.66 | 6.14 ± 0.76 | 5.75 ± 0.49 |
| tPatchGNN | 3.71 ± 0.20 | 3.89 ± 0.16 | 4.56 ± 0.08 | 4.32 ± 0.06 |
| GraFITi | 4.59 ± 0.04 | 4.45 ± 0.04 | 4.91 ± 0.07 | 4.62 ± 0.03 |
| CRU | 4.12 ± 0.08 | 4.43 ± 0.06 | 4.85 ± 0.09 | 4.86 ± 0.07 |
| mTAND | 4.38 ± 0.37 | 4.59 ± 0.29 | 5.29 ± 0.32 | 5.12 ± 0.23 |
| NeuralFlow | 5.47 ± 0.49 | 5.35 ± 0.28 | 6.01 ± 0.91 | 5.66 ± 0.60 |
| Latent-ODE | 5.04 ± 0.46 | 5.11 ± 0.29 | 5.48 ± 0.21 | 5.33 ± 0.14 |
| **Hi-Patch** | **3.29 ± 0.04** | **3.70 ± 0.04** | **4.21 ± 0.08** | **4.25 ± 0.07** |

*Table 6.* Performance of varying observation and forecast horizons on MIMIC-III dataset. The best results are highlighted in **bold**, and the second-best results are in underlined. The results in the table are presented in the form of (Mean ± Std).

| Horizon | 36h → 12h | | 12h → 36h | |
|---|---|---|---|---|
| Metric | MSE$\times 10^{-2}$ | MAE$\times 10^{-2}$ | MSE$\times 10^{-2}$ | MAE$\times 10^{-2}$ |
| iTransformer | 7.27 ± 0.04 | 21.52 ± 0.09 | 7.44 ± 0.21 | 21.47 ± 0.45 |
| ModernTCN | 5.12 ± 0.59 | 11.41 ± 0.30 | 3.96 ± 0.23 | 10.79 ± 0.39 |
| TimesNet | 2.02 ± 0.09 | 8.18 ± 0.22 | 2.39 ± 0.06 | 8.35 ± 0.20 |
| PatchTST | 8.81 ± 0.96 | 22.97 ± 1.32 | 7.11 ± 0.65 | 19.96 ± 1.24 |
| Pathformer | – | – | 2.45 ± 0.02 | 8.65 ± 0.09 |
| TimeMixer | 4.31 ± 0.29 | 13.44 ± 0.28 | 6.31 ± 0.04 | 18.41 ± 0.06 |
| MSGNet | 2.48 ± 0.12 | 9.42 ± 0.32 | 2.57 ± 0.03 | 9.02 ± 0.06 |
| MICN | 2.15 ± 0.13 | 8.66 ± 0.26 | 2.39 ± 0.06 | 8.58 ± 0.11 |
| Warpformer | 1.45 ± 0.10 | 6.74 ± 0.08 | 2.32 ± 0.04 | 8.14 ± 0.07 |
| Raindrop | 2.21 ± 0.37 | 9.17 ± 0.49 | 2.36 ± 0.03 | 8.63 ± 0.11 |
| GRU-D | 2.03 ± 0.13 | 8.14 ± 0.26 | 2.39 ± 0.02 | 8.43 ± 0.13 |
| tPatchGNN | **1.44 ± 0.08** | 6.78 ± 0.14 | 2.35 ± 0.03 | 8.23 ± 0.08 |
| GraFITi | 1.61 ± 0.27 | 7.16 ± 0.36 | **2.22 ± 0.05** | 8.13 ± 0.13 |
| CRU | 2.00 ± 0.13 | 8.16 ± 0.26 | 2.34 ± 0.05 | 8.32 ± 0.13 |
| mTAND | 2.01 ± 0.09 | 8.13 ± 0.23 | 2.29 ± 0.03 | 8.38 ± 0.08 |
| NeuralFlow | 1.97 ± 0.12 | 8.39 ± 0.25 | 2.26 ± 0.08 | 8.29 ± 0.10 |
| Latent-ODE | 1.90 ± 0.03 | 7.92 ± 0.17 | 2.38 ± 0.05 | 8.35 ± 0.13 |
| **Hi-Patch** | 1.56 ± 0.10 | **6.71 ± 0.16** | 2.32 ± 0.02 | **8.11 ± 0.08** |

*Table 7.* Performance of varying observation and forecast horizons on USHCN dataset. The best results are highlighted in **bold**, and the second-best results are in underlined. The results in the table are presented in the form of (Mean ± Std).

| Horizon | 24months → 6months | | 24months → 12months | |
|---|---|---|---|---|
| Metric | MSE×$10^{-1}$ | MAE×$10^{-1}$ | MSE×$10^{-1}$ | MAE×$10^{-1}$ |
| iTransformer | 6.05 ± 0.03 | 3.89 ± 0.14 | 6.19 ± 0.12 | 4.01 ± 0.12 |
| ModernTCN | 6.03 ± 0.10 | 3.68 ± 0.06 | 6.19 ± 0.07 | 3.75 ± 0.03 |
| TimesNet | 5.68 ± 0.05 | 3.66 ± 0.06 | 5.84 ± 0.06 | 3.78 ± 0.10 |
| PatchTST | 6.12 ± 0.03 | 4.01 ± 0.08 | 6.55 ± 0.12 | 4.20 ± 0.06 |
| TimeMixer | 5.89 ± 0.07 | 3.65 ± 0.05 | 6.06 ± 0.27 | 3.76 ± 0.04 |
| MSGNet | 5.76 ± 0.02 | 3.74 ± 0.08 | 5.93 ± 0.05 | 3.77 ± 0.03 |
| MICN | 6.02 ± 0.03 | 3.85 ± 0.04 | 6.00 ± 0.03 | 3.85 ± 0.03 |
| Warpformer | 5.12 ± 0.03 | 3.13 ± 0.08 | 5.10 ± 0.07 | 3.13 ± 0.12 |
| Raindrop | 7.01 ± 0.49 | 4.24 ± 0.33 | 7.61 ± 0.02 | 4.61 ± 0.05 |
| GRU-D | 5.29 ± 0.09 | 3.34 ± 0.09 | 5.36 ± 0.12 | 3.25 ± 0.07 |
| tPatchGNN | 5.23 ± 0.02 | 3.24 ± 0.19 | 6.23 ± 0.10 | 3.83 ± 0.60 |
| GraFITi | 5.12 ± 0.14 | 3.09 ± 0.10 | **5.01 ± 0.03** | 3.14 ± 0.06 |
| CRU | 6.77 ± 1.04 | 4.11 ± 0.61 | 6.64 ± 0.95 | 4.08 ± 0.51 |
| mTAND | 5.16 ± 0.10 | 3.10 ± 0.07 | 5.07 ± 0.03 | 3.09 ± 0.03 |
| NeuralFlow | 5.52 ± 0.05 | 3.46 ± 0.05 | 5.48 ± 0.37 | 3.56 ± 0.37 |
| Latent-ODE | 5.18 ± 0.04 | 3.36 ± 0.04 | 5.23 ± 0.04 | 3.35 ± 0.02 |
| **Hi-Patch** | **5.07 ± 0.17** | **3.06 ± 0.06** | 5.02 ± 0.05 | **3.07 ± 0.08** |

*Table 8.* Performance of varying observation and forecast horizons on PhysioNet dataset. The best results are highlighted in **bold**, and the second-best results are in underlined. The results in the table are presented in the form of (Mean ± Std).

| Horizon | 36h → 12h | | 12h → 36h | |
|---|---|---|---|---|
| Metric | MSE×$10^{-3}$ | MAE×$10^{-2}$ | MSE×$10^{-3}$ | MAE×$10^{-2}$ |
| iTransformer | 56.83 ± 21.17 | 17.05 ± 5.21 | 54.17 ± 17.91 | 17.03 ± 4.18 |
| ModernTCN | 24.87 ± 9.44 | 6.58 ± 0.58 | 31.48 ± 3.96 | 6.73 ± 0.53 |
| TimesNet | 9.43 ± 0.53 | 5.55 ± 0.20 | 9.26 ± 0.18 | 5.58 ± 0.10 |
| PatchTST | 26.13 ± 1.75 | 11.25 ± 0.29 | 25.63 ± 1.51 | 10.70 ± 0.38 |
| Pathformer | 6.85 ± 0.42 | 4.61 ± 0.18 | 8.19 ± 0.28 | 5.03 ± 0.07 |
| TimeMixer | 12.52 ± 0.45 | 6.47 ± 0.21 | 19.86 ± 0.17 | 8.38 ± 0.18 |
| MSGNet | 10.44 ± 0.44 | 5.94 ± 0.08 | 9.91 ± 0.12 | 5.74 ± 0.08 |
| MICN | 10.98 ± 0.31 | 6.01 ± 0.11 | 10.24 ± 0.14 | 5.85 ± 0.06 |
| Warpformer | 4.17 ± 0.13 | 3.38 ± 0.08 | 6.51 ± 0.12 | 4.24 ± 0.04 |
| Raindrop | 10.67 ± 0.33 | 5.87 ± 0.20 | 10.24 ± 0.18 | 5.83 ± 0.10 |
| GRU-D | 6.85 ± 0.37 | 4.88 ± 0.18 | 7.80 ± 0.22 | 5.13 ± 0.13 |
| tPatchGNN | 4.22 ± 0.09 | 3.38 ± 0.04 | 6.45 ± 0.11 | 4.24 ± 0.09 |
| GraFITi | 4.58 ± 0.11 | 3.65 ± 0.05 | 6.30 ± 0.14 | 4.38 ± 0.12 |
| CRU | 6.74 ± 0.21 | 4.82 ± 0.11 | 7.66 ± 0.14 | 4.97 ± 0.05 |
| mTAND | 5.61 ± 0.31 | 4.15 ± 0.09 | 7.46 ± 0.19 | 4.85 ± 0.05 |
| NeuralFlow | 8.87 ± 1.00 | 5.43 ± 0.18 | 7.98 ± 0.57 | 5.08 ± 0.24 |
| Latent-ODE | 6.99 ± 0.24 | 4.74 ± 0.11 | 7.28 ± 0.13 | 4.83 ± 0.07 |
| **Hi-Patch** | **4.16 ± 0.08** | **3.31 ± 0.06** | **6.30 ± 0.06** | **4.12 ± 0.05** |

## F.2. Leave-variables-out Classification

*Table 9.* Classification performance on samples with a fixed set of left-out variables. The best results are highlighted in **bold** and the second best results are in underlined.

| Dataset | Methods | Discard ratio | | | | | | | | | |
|---|---|---|---|---|---|---|---|---|---|---|---|
| | | 10% | | 20% | | 30% | | 40% | | 50% | |
| | | AUROC | AUPRC | AUROC | AUPRC | AUROC | AUPRC | AUROC | AUPRC | AUROC | AUPRC |
| P12 | GRU-D | 68.6 ± 2.3 | 35.8 ± 2.2 | 68.2 ± 2.1 | 34.5 ± 2.9 | 66.8 ± 3.3 | 32.7 ± 4.6 | 65.8 ± 4.0 | 31.3 ± 5.2 | 65.1 ± 4.1 | 30.4 ± 5.5 |
| | mTAND | 74.9 ± 0.6 | 37.7 ± 0.6 | 74.0 ± 1.3 | 36.5 ± 1.5 | 71.4 ± 3.8 | 34.1 ± 3.7 | 70.6 ± 3.6 | 33.2 ± 3.7 | 70.1 ± 3.5 | 32.5 ± 3.6 |
| | DGM$^2$-O | 76.3 ± 1.1 | 39.3 ± 1.5 | 76.1 ± 1.1 | 38.2 ± 1.7 | 74.8 ± 2.2 | 36.8 ± 2.6 | 72.0 ± 5.3 | 34.3 ± 5.0 | 70.4 ± 5.9 | 32.7 ± 5.7 |
| | MTGNN | 71.2 ± 2.1 | 30.5 ± 1.5 | 70.3 ± 3.3 | 29.7 ± 2.8 | 68.9 ± 4.2 | 28.5 ± 3.3 | 68.1 ± 4.7 | 27.7 ± 3.6 | 67.6 ± 5.2 | 27.2 ± 3.8 |
| | Raindrop | 73.2 ± 1.6 | 32.4 ± 0.9 | 73.0 ± 1.6 | 31.7 ± 1.1 | 72.2 ± 2.6 | 31.1 ± 2.7 | 71.5 ± 3.5 | 30.6 ± 3.5 | 70.8 ± 4.2 | 29.7 ± 4.3 |
| | StraTS | **80.8 ± 0.4** | 42.4 ± 1.8 | **80.4 ± 0.7** | **41.8 ± 1.8** | **79.5 ± 1.6** | **40.2 ± 3.1** | 78.7 ± 1.9 | 38.4 ± 4.2 | 78.4 ± 2.0 | 37.5 ± 4.4 |
| | DuETT | 73.9 ± 1.7 | 35.8 ± 2.3 | 74.7 ± 1.8 | 35.3 ± 2.0 | 73.6 ± 2.2 | 34.1 ± 2.4 | 72.8 ± 2.6 | 33.3 ± 2.7 | 72.3 ± 2.7 | 32.6 ± 2.8 |
| | Warpformer | 75.9 ± 0.7 | 37.3 ± 2.2 | 75.6 ± 0.8 | 36.7 ± 2.3 | 73.8 ± 2.9 | 34.3 ± 4.1 | 72.8 ± 3.4 | 33.0 ± 4.6 | 72.1 ± 3.7 | 32.2 ± 4.7 |
| | **Hi-Patch** | 80.1 ± 1.1 | **43.3 ± 2.2** | 79.7 ± 1.2 | **41.8 ± 2.7** | 79.1 ± 1.5 | **40.2 ± 3.4** | **78.8 ± 1.5** | **39.6 ± 3.2** | **78.7 ± 1.4** | **39.4 ± 3.0** |
| P19 | GRU-D | 88.5 ± 2.3 | 54.6 ± 3.7 | 88.8 ± 2.1 | 54.2 ± 3.4 | 88.0 ± 2.5 | 50.4 ± 7.5 | 87.5 ± 2.8 | 49.6 ± 6.9 | 86.4 ± 3.5 | 47.2 ± 8.6 |
| | mTAND | 79.6 ± 1.8 | 28.6 ± 1.9 | 79.2 ± 1.9 | 28.1 ± 2.1 | 78.0 ± 2.4 | 26.9 ± 2.9 | 77.2 ± 2.7 | 26.3 ± 2.9 | 76.2 ± 3.2 | 24.3 ± 4.8 |
| | DGM$^2$-O | 87.4 ± 0.6 | 53.4 ± 1.5 | 87.3 ± 0.8 | 53.2 ± 1.7 | 86.6 ± 1.6 | 49.9 ± 5.1 | 85.8 ± 1.9 | 47.7 ± 5.9 | 85.2 ± 2.2 | 45.7 ± 6.7 |
| | MTGNN | 84.5 ± 1.4 | 48.9 ± 2.3 | 84.8 ± 1.7 | 49.8 ± 3.1 | 84.0 ± 1.9 | 47.2 ± 4.8 | 83.3 ± 2.2 | 45.5 ± 5.5 | 82.5 ± 2.9 | 42.7 ± 9.2 |
| | Raindrop | 88.2 ± 1.5 | 59.7 ± 1.5 | 88.1 ± 1.3 | 59.8 ± 1.4 | 87.8 ± 1.2 | 59.1 ± 1.7 | 87.6 ± 1.1 | 58.5 ± 1.9 | 87.1 ± 1.5 | 57.7 ± 2.3 |
| | StraTS | 90.6 ± 0.9 | 56.4 ± 3.0 | 91.0 ± 0.9 | 56.3 ± 2.3 | 91.0 ± 0.9 | 56.0 ± 2.4 | 90.8 ± 1.0 | 55.1 ± 3.0 | 90.4 ± 1.3 | 54.4 ± 3.3 |
| | DuETT | 85.2 ± 1.0 | 53.7 ± 1.0 | 84.8 ± 1.1 | 53.9 ± 0.8 | 84.7 ± 1.0 | 53.3 ± 1.6 | 84.3 ± 1.4 | 52.7 ± 2.1 | 84.4 ± 1.3 | 52.5 ± 2.0 |
| | Warpformer | 91.3 ± 0.8 | 55.2 ± 5.6 | 91.3 ± 0.8 | 55.1 ± 5.6 | 91.4 ± 0.8 | 56.0 ± 4.8 | 91.5 ± 0.7 | 56.4 ± 4.3 | 91.2 ± 0.8 | 56.2 ± 3.9 |
| | **Hi-Patch** | **92.1 ± 0.4** | **60.7 ± 2.0** | **92.0 ± 0.4** | **60.6 ± 1.9** | **91.9 ± 0.5** | **60.3 ± 1.8** | **91.9 ± 0.5** | **60.0 ± 1.7** | **91.6 ± 0.8** | **59.5 ± 1.9** |
| PhysioNet | GRU-D | 70.0 ± 3.0 | 32.1 ± 4.1 | 69.5 ± 2.6 | 31.1 ± 3.6 | 69.2 ± 3.0 | 31.0 ± 4.4 | 68.3 ± 3.6 | 30.1 ± 5.3 | 68.1 ± 3.7 | 29.8 ± 5.3 |
| | mTAND | 80.5 ± 2.1 | **42.8 ± 4.0** | 78.2 ± 3.4 | 40.5 ± 4.7 | 76.3 ± 4.0 | 37.7 ± 5.7 | 75.6 ± 3.9 | 36.6 ± 5.6 | 75.1 ± 3.9 | 36.1 ± 5.1 |
| | DGM$^2$-O | 80.2 ± 0.9 | 38.6 ± 2.8 | 80.4 ± 0.9 | 38.3 ± 2.8 | 79.3 ± 1.9 | 37.1 ± 3.4 | 77.5 ± 3.7 | 35.4 ± 4.4 | 75.6 ± 5.0 | 34.0 ± 4.8 |
| | MTGNN | 68.9 ± 4.1 | 25.8 ± 4.8 | 69.3 ± 4.3 | 26.6 ± 4.5 | 69.0 ± 4.8 | 26.3 ± 5.2 | 68.3 ± 5.2 | 25.4 ± 4.8 | 67.2 ± 5.4 | 24.4 ± 4.8 |
| | Raindrop | 76.5 ± 1.2 | 33.4 ± 2.2 | 76.5 ± 1.3 | 32.3 ± 2.3 | 75.6 ± 2.0 | 30.8 ± 3.2 | 74.7 ± 2.6 | 29.7 ± 3.5 | 73.6 ± 3.2 | 28.8 ± 3.9 |
| | StraTS | 80.4 ± 2.4 | 40.8 ± 2.6 | **80.8 ± 2.3** | 40.5 ± 2.2 | 79.5 ± 2.8 | 39.5 ± 2.7 | 78.8 ± 3.0 | 38.2 ± 3.8 | 78.1 ± 3.4 | 37.6 ± 3.7 |
| | DuETT | 78.2 ± 2.8 | 39.9 ± 3.5 | 78.3 ± 3.0 | 39.9 ± 3.7 | 76.7 ± 3.7 | 37.9 ± 4.5 | 75.9 ± 3.8 | 37.0 ± 4.6 | 74.9 ± 4.3 | 35.9 ± 5.0 |
| | Warpformer | 78.2 ± 1.0 | 33.3 ± 2.1 | 77.7 ± 1.6 | 33.6 ± 1.8 | 75.8 ± 3.4 | 31.8 ± 3.0 | 73.8 ± 4.6 | 30.2 ± 4.1 | 72.7 ± 4.9 | 29.2 ± 4.2 |
| | **Hi-Patch** | **81.2 ± 3.4** | 42.0 ± 7.6 | 80.6 ± 3.4 | **41.2 ± 7.2** | **79.8 ± 3.4** | **40.4 ± 6.3** | **79.3 ± 3.3** | **39.8 ± 5.8** | **78.7 ± 3.4** | **39.3 ± 5.5** |
| MIMIC-III | GRU-D | 81.0 ± 0.6 | 42.1 ± 0.8 | 80.3 ± 0.9 | 41.7 ± 1.0 | 79.2 ± 1.8 | 41.0 ± 1.4 | 78.5 ± 2.1 | 40.4 ± 1.6 | 77.9 ± 2.2 | 39.9 ± 1.8 |
| | mTAND | 81.2 ± 0.2 | 42.1 ± 0.8 | 80.4 ± 1.1 | 41.9 ± 1.2 | 79.7 ± 1.4 | 41.0 ± 1.7 | 79.3 ± 1.4 | 40.4 ± 2.0 | 78.8 ± 1.6 | 39.8 ± 2.3 |
| | DGM$^2$-O | 78.8 ± 0.5 | 34.2 ± 0.9 | 78.3 ± 0.8 | 33.9 ± 1.1 | 77.6 ± 1.2 | 33.4 ± 1.2 | 77.3 ± 1.3 | 33.1 ± 1.2 | 76.8 ± 1.5 | 32.6 ± 1.4 |
| | MTGNN | 78.8 ± 1.1 | 34.5 ± 1.4 | 78.0 ± 1.6 | 34.0 ± 1.3 | 77.1 ± 2.2 | 33.5 ± 1.5 | 76.3 ± 2.5 | 32.8 ± 1.9 | 75.6 ± 3.2 | 32.2 ± 2.4 |
| | Raindrop | 78.2 ± 1.1 | 33.7 ± 0.9 | 77.5 ± 1.3 | 33.5 ± 0.9 | 76.4 ± 2.1 | 32.8 ± 1.4 | 76.0 ± 2.0 | 32.5 ± 1.4 | 75.7 ± 2.0 | 32.3 ± 1.4 |
| | StraTS | 82.4 ± 0.7 | 43.3 ± 2.8 | 82.1 ± 0.6 | 43.7 ± 2.1 | 81.7 ± 0.8 | 43.1 ± 2.0 | **81.5 ± 0.8** | 42.6 ± 2.0 | **81.0 ± 1.3** | 41.9 ± 2.4 |
| | DuETT | 78.0 ± 0.5 | 34.0 ± 0.9 | 77.2 ± 1.0 | 33.7 ± 0.8 | 76.6 ± 1.2 | 33.3 ± 1.0 | 76.4 ± 1.2 | 33.0 ± 1.0 | 76.1 ± 1.3 | 32.6 ± 1.3 |
| | Warpformer | 82.5 ± 0.5 | 43.1 ± 0.8 | 81.7 ± 0.9 | 42.5 ± 1.2 | 81.2 ± 1.1 | 42.1 ± 1.2 | 80.6 ± 1.5 | 41.8 ± 1.3 | 80.0 ± 1.9 | 41.3 ± 1.6 |
| | **Hi-Patch** | **82.8 ± 0.3** | **44.3 ± 1.2** | **82.3 ± 0.6** | **44.1 ± 1.3** | **81.7 ± 1.0** | **43.6 ± 1.4** | 81.1 ± 1.5 | **42.8 ± 2.2** | 80.5 ± 1.9 | **42.0 ± 2.6** |

## F.3. Regular Multivariate Time Series Forecasting

We test Hi-Patch on traditional datasets with regular sampling frequency to enhance the model's generality. Specifically, we add forecasting experiments on four regular MTS datasets: ETTm1, ETTm2, Exchange and Weather. The lengths of the historical and prediction windows are both set to 96. The experimental pipeline is based on that provided by iTransformer (Liu et al., 2024), and we compared our model with five SOTA regular time series forecasting methods in recent years: PatchTST (Nie et al., 2023), Crossformer (Zhang & Yan, 2023), FEDformer (Zhou et al., 2022) and Autoformer (Wu et al., 2021a).

Table 10 shows the results. From the experimental results, our Hi-Patch achieved an average ranking of 2.25 in terms of MSE and MAE metrics across the four datasets, notably outperforming the latest iTransformer on the Exchange and Weather datasets. Although our method is initially designed for IMTS, its ability to extract multi-scale temporal dependencies and inter-variable dependencies is general for time series modeling. Consequently, it can also achieve competitive performance in regular time series forecasting tasks, further demonstrating the generality of our method.

*Table 10.* Forecasting performance of Hi-Patch on four regular time series datasets.

| Methods | ETTm1 | | ETTm2 | | Exchange | | Weather | |
|---|---|---|---|---|---|---|---|---|
| | MSE | MAE | MSE | MAE | MSE | MAE | MSE | MAE |
| iTransformer | 0.334 | 0.368 | 0.180 | 0.264 | 0.086 | 0.206 | 0.174 | 0.214 |
| PatchTST | 0.329 | 0.367 | 0.175 | 0.259 | 0.088 | 0.205 | 0.177 | 0.218 |
| Crossformer | 0.404 | 0.426 | 0.287 | 0.366 | 0.256 | 0.367 | 0.158 | 0.230 |
| FEDformer | 0.379 | 0.419 | 0.203 | 0.287 | 0.148 | 0.278 | 0.217 | 0.296 |
| Autoformer | 0.505 | 0.475 | 0.203 | 0.287 | 0.197 | 0.323 | 0.266 | 0.336 |
| **Hi-Patch** | 0.380(4th) | 0.405(3rd) | 0.184(3rd) | 0.269(3rd) | 0.085(1st) | 0.206(2nd) | 0.158(1st) | 0.208(1st) |

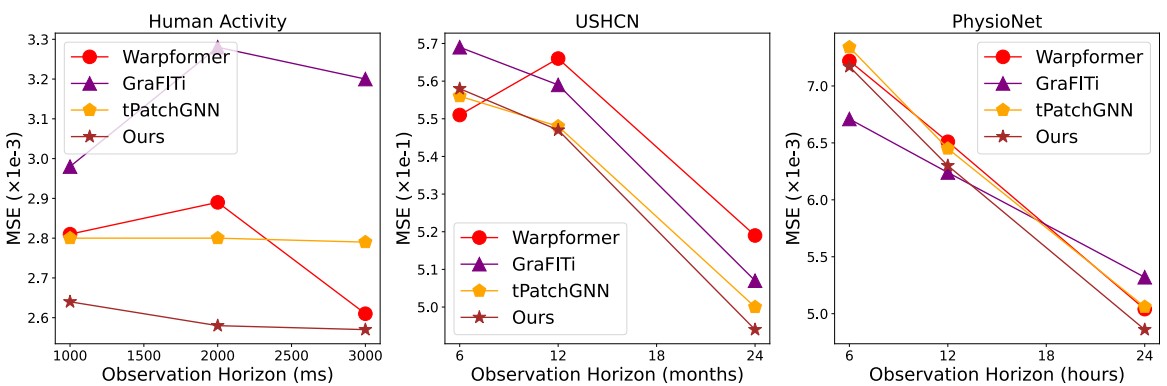

*Figure 6.* Forecasting performance with varing observation horizons and fixed forecast horizon.

## F.4. Varying Observation Horizons and Fixed Forecast Horizon

We conduct additional experiments to evaluate the model's ability to capture more temporal dependencies from long-term data on 3 datasets. Specifically:

**Human Activity**   Fixed forecast length at 1000ms, varied observation horizons as {1000, 2000, 3000}ms.

**USHCN**   Fixed forecast length at 1 month, varied observation horizons as {6, 12, 24}months.

**PhysioNet**   Fixed forecast length at 24hours, varied observation horizons as {6, 12, 24}hours.

Figure 6 shows the results of our model and the three most competitive baseline methods: Warpformer, GraFITi and tPatchGNN. It is evident that our method consistently improves in MSE as the historical length increases. This demonstrates

the model's capability to learn more temporal dependencies from longer-term data. Conversely, some existing IMTS methods do not have this ability, such as Warpformer on the USHCN and Human Activity datasets, and GraFITi on the Human Activity dataset, both showing an increase in MSE with increasing observation horizons. These supplementary experiments further highlight the superiority of our Hi-Patch approach compared to prior methods.

## G. Computational Complexity Analysis

### G.1. Theoretical Analysis

For a dataset with $V$ variables, maximum series length $L_{\max}$, and missing rate $M$, suppose the data is divided into $N$ patches using a fixed time span. Hi-Patch consists of one Intra-Patch Graph Layer and $\log_2 N$ Inter-Patch Graph Layers. We analyze the computational complexity of these two components separately.

**Intra-Patch Graph Layer Complexity**    For each patch, assuming missing points are uniformly distributed, the number of nodes per patch is $VL_{\max}(1-M)/N$. The complexity of attention operations within a patch is $\mathcal{O}((VL_{\max}(1-M)/N)^2)$. For $N$ patches computed in parallel, the total complexity is $\mathcal{O}(V^2L_{\max}^2(1-M)^2/N)$. Additionally, aggregating nodes of the same variable to the reference time point has a complexity of $\mathcal{O}(VL_{\max}N(1-M))$. Thus, the overall complexity of the Intra-Patch Graph Layer is $\mathcal{O}(V^2L_{\max}^2(1-M)^2/N + VL_{\max}N(1-M)) = \mathcal{O}(V^2L_{\max}^2(1-M)^2/N)$.

**Inter-Patch Graph Layers Complexity**    In the first Inter-Patch Graph Layer, the maximum number of nodes is $NV$. Each node computes graph attention with all $2V$ variable nodes in adjacent patches, resulting in a complexity of $\mathcal{O}(2NV^2)$. Aggregation, where each node interacts with only one adjacent node of the same variable, has a complexity of $\mathcal{O}(NV)$. Thus, the first Inter-Patch Graph Layer has a total complexity of $\mathcal{O}(2NV^2 + NV) = \mathcal{O}(NV^2)$. For the second layer, the number of nodes is halved compared to the first layer, leading to a complexity of $\mathcal{O}(NV^2/2)$. Summing over $\log_2 N$ layers, the total complexity of Inter-Patch Graph Layers is $\mathcal{O}\left(\sum_{k=1}^{\log_2 N} \frac{NV^2}{2^{k-1}}\right) = \mathcal{O}(V^2(2N-2))$.

From the above, the primary complexity of our model lies in the first Intra-Patch Graph Layer. Since each observation is treated as a node for attention operations to capture asynchronous dependencies in IMTS, this layer has quadratic complexity with respect to both the number of variables and the sequence length. Fortunately, IMTS datasets are often highly sparse (missing rate $M > 80\%$ as shown in Table 4), and the coefficient $(1-M)^2$ reduces the practical complexity by 1–2 orders of magnitude. Additionally, increasing the number of patches $N$ further reduces computational overhead in practice. Therefore, the practical computation time of our method is acceptable, as shown in Appendix G.2. In summary, our method achieves a more comprehensive extraction of multi-scale asynchronous correlations in IMTS at the cost of an acceptable increase in computational complexity.

### G.2. Empirical Analysis

We compared 8 leading IMTS models: Warpformer, GRU-D, GraFITi, tPatchGNN, CRU, mTAND, NeuralFlow, and Latent-ODE. All models are evaluated using the same batch size (32 for Human Activity, 128 for USHCN, 64 for PhysioNet, and 8 for MIMIC-III) to assess their training time per epoch and MSE. The results shown in Figure 7 indicate that the training time of our method ranks 5-th on average across the four datasets. It is worth noting that our model leverages a hierarchical graph structure to extract asynchronous correlations and multi-scale features in IMTS that other methods fail to capture, which inevitably leads to an increase in computational overhead. Nonetheless, our model's training time remains in the same order of magnitude as the fastest models. In this context, We believe that the trade-off of sacrificing some training time to extract richer features in IMTS is worthwhile.

### G.3. Future Scalability on Dense Data

Should the need arise to extend our approach to large-scale dense datasets, existing scalability techniques can be integrated to reduce the overhead of the intra-patch graph. For instance, edge pruning—such as connecting each node only to its nearest $k$ observation points—can reduce the complexity to $\mathcal{O}(kN)$, or alternatively, leveraging improved attention mechanisms like the ProbSparse Self-Attention from Informer (Zhou et al., 2021) can lower the complexity to $\mathcal{O}(N \log N)$. In these cases, the trade-off between a slight reduction in feature extraction and enhanced scalability would be worthwhile.

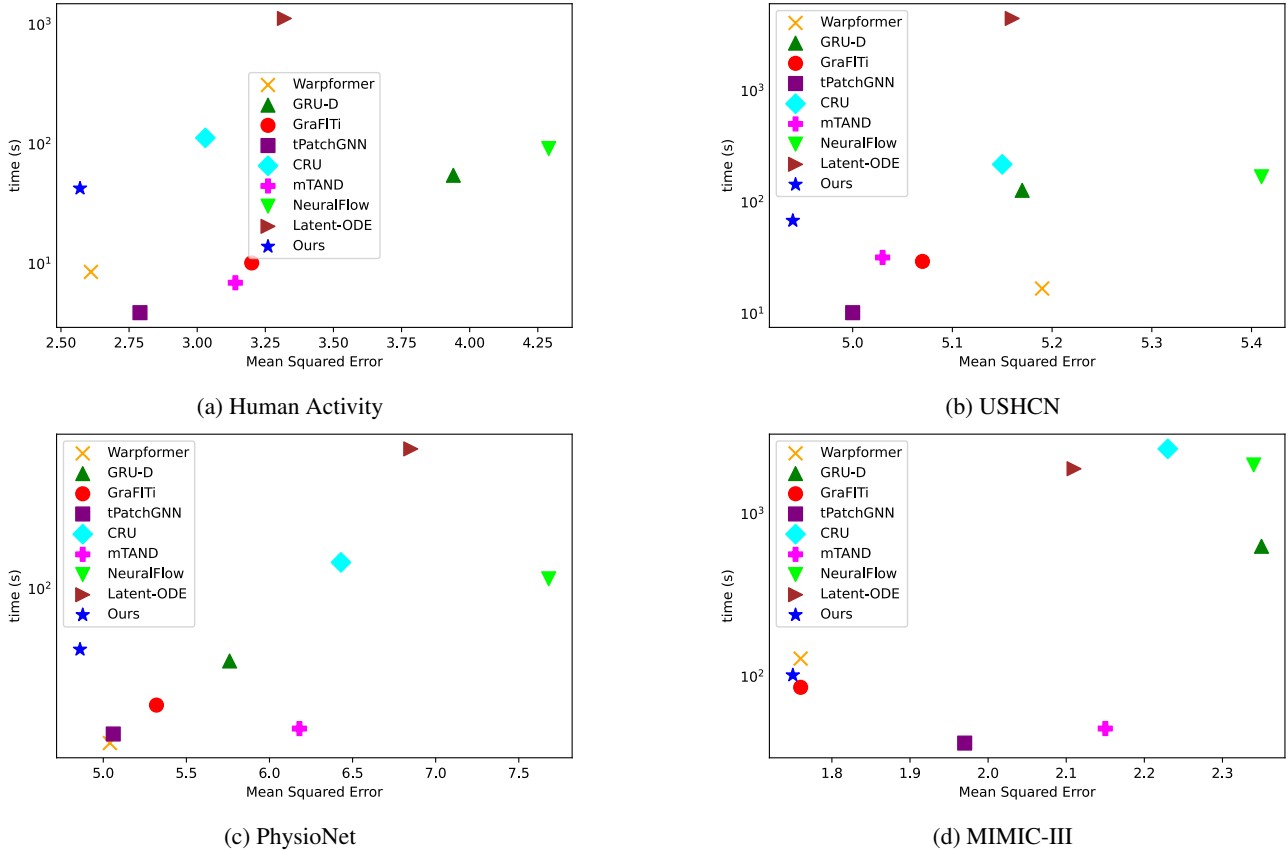

(a) Human Activity

(b) USHCN

(c) PhysioNet

(d) MIMIC-III

*Figure 7.* Comparison of IMTS models in terms of efficiency: training time per epoch against error metric.

## H. Comprehensive Discussion on the Technical Details of Hi-Patch and Existing Methods

**Differences from patch-based methods for regular multivariate time series**  Representative methods include PatchTST (Nie et al., 2023) and Pathformer (Chen et al., 2024a). These approaches encode observations within fixed-length time steps of univariate time series into patches and use attention mechanisms to extract temporal correlations between patches. However, these methods apply a uniform time step to all variables, which makes them unsuitable for IMTS. In IMTS, some variables may only have a few observations, resulting in patches with long time spans, while other variables with frequent observations will result in patches covering very short time spans. These methods cannot account for differences in time spans across patches, and the patches for different variables are often misaligned temporally, complicating the extraction of inter-variable correlations. In contrast, Hi-Patch segments patches using fixed time spans and each patch contains observations for all variables. This design guarantees consistent time spans across patches and facilitates the modeling of inter-variable relationships.

**Differences from GNN-based methods for regular multivariate time series**  A latest typical example is MSGNet (Cai et al., 2024), which extracts correlations between variables at the same timestamp using variable graphs. However, such methods struggle to capture asynchronous correlations that are prevalent in IMTS. Our method uses both intra-patch and inter-patch graphs to simultaneously extract: 1) Correlations of the same variable across different time points (SVDT), 2) Correlations of different variables at the same time point (DVST) and 3) Correlations of different variables across different time points (DVDT).

**Differences from patch-based methods for irregular multivariate time series**  Pioneering work is tPatchGNN (Zhang et al., 2024), which also segments patches based on fixed time spans and models inter-variable correlations via GNNs. However, tPatchGNN considers only one variable per patch and fails to capture fine-grained inter-variable correlations within a patch. Moreover, it extracts features separately in temporal and variable dimensions, limiting its ability to model

asynchronous correlations. In contrast, Hi-Patch extracts fine-grained dependencies within patches and coarse-grained dependencies across patches while modeling both synchronous and asynchronous inter-variable correlations. Additionally, Hi-Patch can extract multi-scale information from IMTS, which is absent in tPatchGNN.

**Differences from GNN-based methods for irregular multivariate time series** Representative methods include Raindrop (Zhang et al., 2022) and GraFITi (Yalavarthi et al., 2024). Raindrop models variable interactions at each timestamp using GNNs but cannot capture asynchronous correlations. GraFITi represents IMTS using a variable-time bipartite graph but, as acknowledged in their paper, has limited ability to extract asynchronous correlations. Moreover, these methods lack the capability to extract multi-scale information from IMTS. Hi-Patch stands out by capturing both synchronous and asynchronous dependencies at multiple scales, which provides a significant advantage over these methods.

**Differences from multi-scale methods for irregular multivariate time series** Among methods specifically designed for IMTS, Warpformer (Zhang et al., 2023) is the only method that considers the multi-scale characteristics of IMTS and it achieves the most competitive performance. However, Warpformer involves interpolation of sparse variables, which may distort the original distribution of IMTS. In addition, Warpformer employs attention separately in the time and variable dimensions, limiting its ability to capture asynchronous dependencies. In contrast, our Hi-Patch flexibly represents and extracts both synchronous and asynchronous dependencies of IMTS under different scales through an intra-patch graph layer and several inter-patch graph layers. Moreover, Hi-Patch only handles actually observed points, avoiding the accumulation of imputation errors, particularly in cases of higher missing ratios, exhibiting a degree of robustness.

## I. More Results for Ablation Study

*Table 11.* Ablation results of Hi-Patch on two datasets. The results in the table are presented in the form of (Mean ± Std).

| Methods | PhysioNet | | MIMIC-III | |
|---|---|---|---|---|
| | MSE$\times 10^{-3}$ | MAE$\times 10^{-2}$ | MSE$\times 10^{-2}$ | MAE$\times 10^{-2}$ |
| **Hi-Patch** | **4.86 ± 0.03** | **3.62 ± 0.07** | **1.75 ± 0.26** | **7.24 ± 0.18** |
| w/o Hie | 4.96 ± 0.07 | 3.68 ± 0.08 | 1.77 ± 0.30 | 7.30 ± 0.13 |
| w/o DVDT | 4.88 ± 0.05 | 3.67 ± 0.04 | 1.80 ± 0.04 | 7.41 ± 0.18 |
| w/o 3W | 4.98 ± 0.07 | 3.74 ± 0.04 | 1.79 ± 0.04 | 7.38 ± 0.09 |
| w/o TEAGG | 6.34 ± 0.60 | 4.44 ± 0.36 | 1.92 ± 0.07 | 7.63 ± 0.26 |

## J. More Results for Effect of Scale Quantity

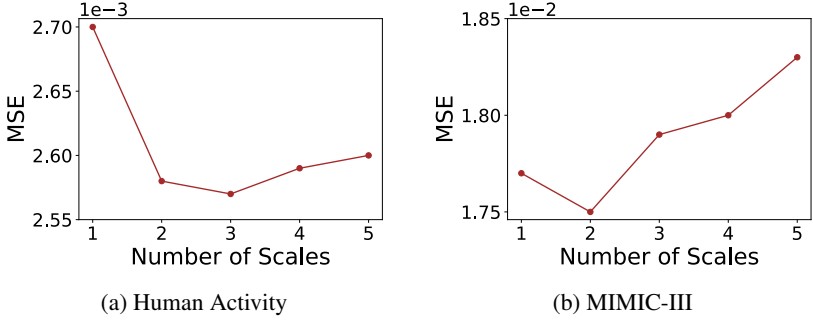

(a) Human Activity        (b) MIMIC-III

*Figure 8.* Effect of different scale quantities on Human Activity and MIMIC-III datasets.

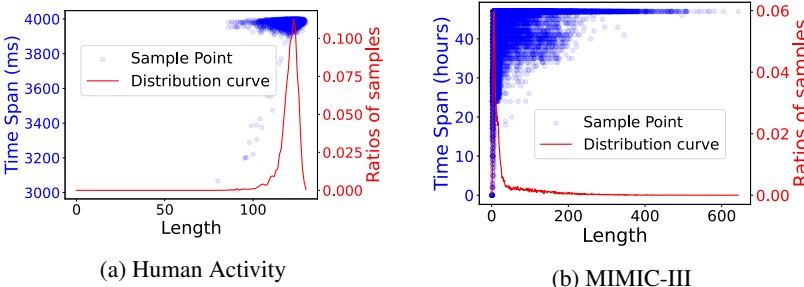

(a) Human Activity

(b) MIMIC-III

*Figure 9.* Distribution of sample length and time span on two datasets. Each blue dot represents a sample, with its x-coordinate indicating the sample length, and the left y-axis representing its time span. The red curve is the distribution curve of sample length.

## K. Visualization of Multi-scale Views

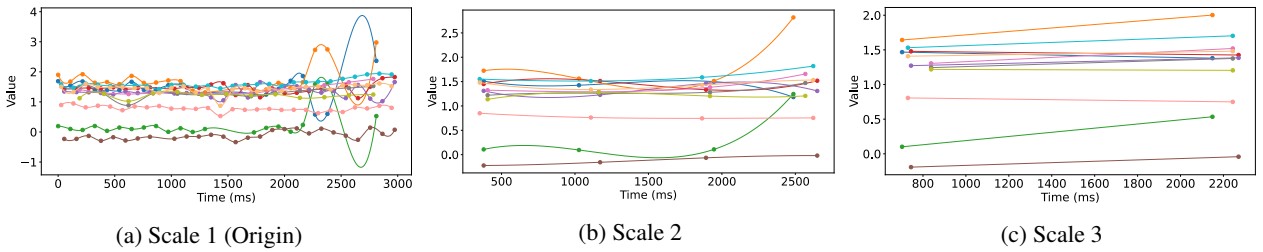

(a) Scale 1 (Origin)

(b) Scale 2

(c) Scale 3

*Figure 10.* Visualization of views on three scales on Human Activity dataset.

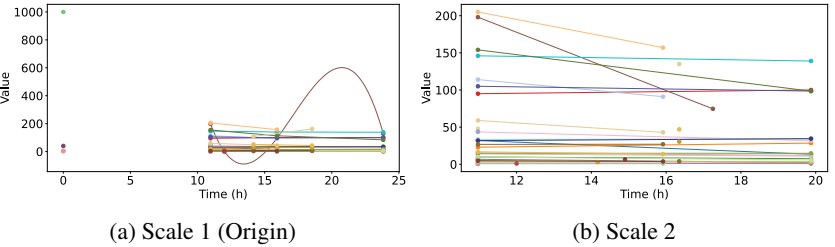

(a) Scale 1 (Origin)

(b) Scale 2

*Figure 11.* Visualization of views on two scales on MIMIC-III dataset.

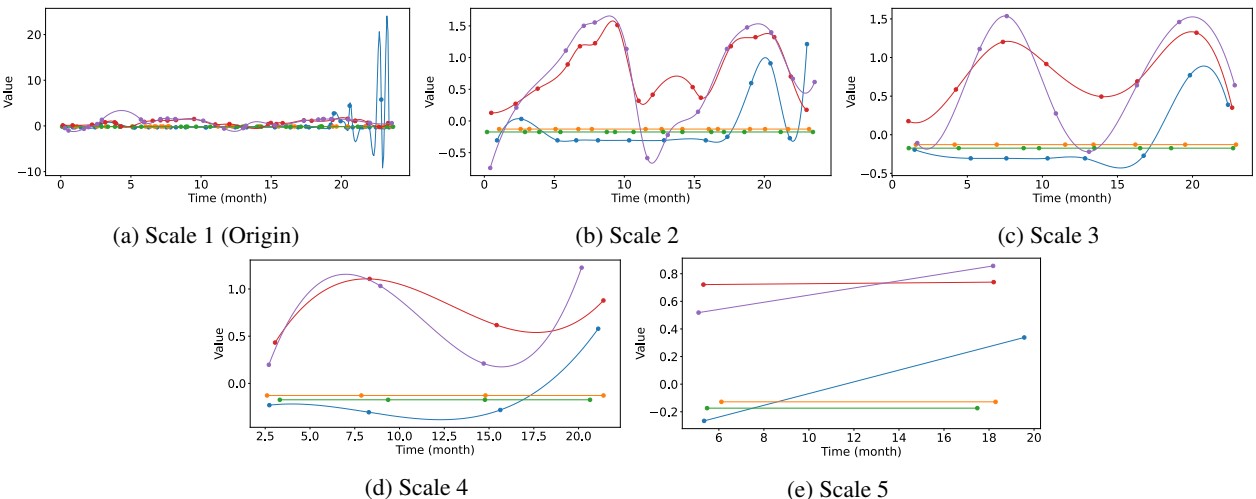

*Figure 12.* Visualization of views on five scales on USHCN dataset.

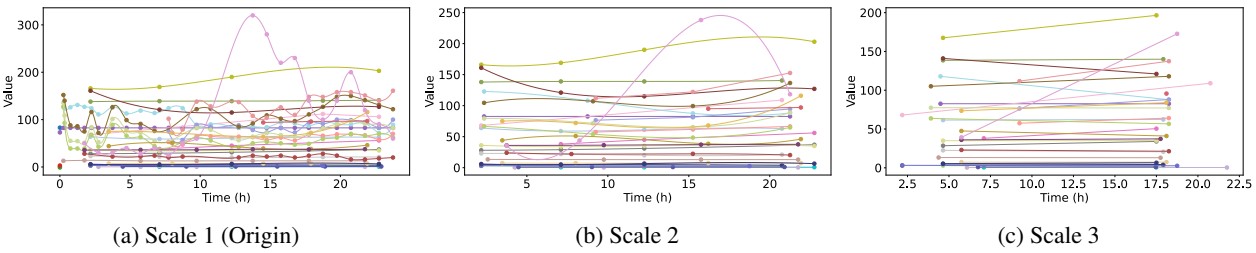

*Figure 13.* Visualization of views on three scales on PhysioNet dataset.

