# OpenReview forum: "Hi-Patch: Hierarchical Patch GNN for Irregular Multivariate Time Series"
_ICML.cc/2025/Conference — ICML 2025 poster_

### Official Review · Reviewer_Ko9v · 2025-03-04

**Overall Recommendation:** 3

**Summary:**

The paper introduces Hi-Patch, a hierarchical patch graph network designed for IMTS, where variables have different sampling rates. Hi-Patch models both local and global dependencies across different scales. It represents observations as nodes, captures short-term dependencies using intra-patch graphs, and progressively learns global features through inter-patch layers. The final representations are fed into task-specific decoders. Experiments on eight datasets show that Hi-Patch outperforms SOTA models in IMTS forecasting and classification.

**Claims And Evidence:**

There are no problematic main claims.

**Essential References Not Discussed:**

No.

**Experimental Designs Or Analyses:**

The experiments in this paper are relatively comprehensive and demonstrate the effectiveness of the model. However, the ablation study does not sufficiently highlight the importance of hierarchical information. Additionally, the hierarchical illustrations in the appendix (Figures 11–13) fail to clearly show that the proposed model effectively captures better hierarchical information.

**Methods And Evaluation Criteria:**

The model designed in the paper is both reasonable and effective for addressing the IMTS modeling problem. The dataset used is a commonly employed real-world dataset in IMTS modeling, which also provides practical guidance for solving real-world problems.

**Other Comments Or Suggestions:**

No.

**Other Strengths And Weaknesses:**

1. The motivation of this paper is unclear. While it emphasizes the importance of multi-scale information in multivariate time series modeling, there is no direct evidence supporting this claim. Even in the ablation study, no dedicated experiments demonstrate the significance of hierarchical information. Additionally, the hierarchical illustrations in the appendix (Figures 11–13) do not convincingly show that the proposed model captures better hierarchical structures.
2. The proposed model involves multiple graph layers, which may impact its time and space complexity. Therefore, it would be helpful if the authors could provide an analysis of Hi-Patch’s computational complexity.
3. The paper requires tuning of several important parameters, including but not limited to the number of layers and patch size.

**Questions For Authors:**

What are the advantages of designing hierarchy at the feature level compared to designing it at the raw value level? Specifically, as shown in the paper, the hierarchical structure obtained through aggregation at the feature level may lead to inappropriate information being passed to the next layer, potentially affecting the learning process in higher layers.

**Relation To Broader Scientific Literature:**

This paper details efforts to advance IMTS modeling across various scientific domains. The paper provides a new perspective on the modeling of ISMTS to some extent and effectively improves downstream task performance.

**Theoretical Claims:**

There are no theoretical claims in this paper.

---

> ### Author Rebuttal · Authors · 2025-03-30
>
> # Responses to Reviewer Ko9v
>
> **Q1. The motivation and experimental validation of IMTS multi-scale modeling.**
>
> **A1.**
>
> 1. **Motivation and Significance of Multi-scale Information**
>
>    Our work is grounded in the premise that multi-scale information is essential for general time series analysis—as demonstrated by previous studies (e.g., Pathformer, TimeMixer, MSGNet).
>
>    Since irregular multivariate time series (IMTS) inherently exhibit multi-scale features (for instance, monthly, quarterly, and yearly patterns in the USHCN weather dataset), it is natural to assume that multi-scale information is also critical for IMTS.
>
>    Moreover, prior work such as Warpformer has empirically verified the benefits of multi-scale modeling for IMTS, while also revealing certain limitations.
>
>    These intuitive and empirical insights jointly motivate our further exploration into multi-scale modeling for IMTS.
>
> 2. **Experimental Validation and Ablation Studies**
>
>    In our ablation experiments (the “w/o Hie” setting), we set the patch size to span the entire historical window, thereby removing the hierarchical structure so that the model processes only single-scale raw information(Table 3 and Table 11 in Appendix I).
>
>    The significant performance drop observed (3.90%↓ on MSE and 3.07%↓ on MAE on four datasets) directly confirms the pivotal role of the hierarchical design in extracting multi-scale features.
>
>    Additionally, Figures 11–13 in Appendix K visually demonstrate the diverse patterns that Hi-Patch can "see" at different layers—a capability that distinguishes our model from many existing IMTS methods.
>
> ------
>
> **Q2. Analysis of computational complexity.**
>
> **A2.** As detailed in Appendix G.1, the primary computational cost of our method arises from the intra-patch graph layer, whose complexity scales quadratically with the number of observation points (N) per patch.
>
> Our focus is on irregular multivariate time series, which are typically characterized by extremely sparse sampling. As shown in Table 4, all our datasets exhibit a missing rate exceeding **75%**. In these scenarios, the computational cost decreases sharply due to the limited number of available data points.
>
> Our empirical study in Appendix G.2 indicates that our overhead is moderate—**ranking 5th out of 9 methods**—and remains within an acceptable range. Consequently, We contend that the additional overhead is justified by the improved extraction of scarce data patterns.
>
> ------
>
> **Q3. Clarification of parameter tuning.**
>
> **A3.**
>
> 1. **Intuitive Parameter Choice**
>    Key parameters—such as the number of layers and patch size—have clear physical interpretations. Layers correspond to different scales of feature extraction, while the patch size reflects the local time window. These parameters can be set according to the specific task requirements and data characteristics. Our sensitivity analyses (Sections 5.4 and 5.5) demonstrate that the optimal parameter ranges is related to the sample distributions and inherent periodicities. Detailed search ranges are provided in Appendix E to guide practical parameter selection.
> 2. **Practicality and Generalizability**
>    Although some hyperparameter tuning is required—common in deep learning methods[1, 2, 3]—our approach shows competitive or superior performance compared to state-of-the-art methods. The tuning process further adds flexibility, allowing the model to adapt to a variety of applications.
>
> [1]. Random Search for Hyper-Parameter Optimization*. Journal of Machine Learning Research, 13:281–305, 2012.
>
> [2]. Practical Bayesian Optimization of Machine Learning Algorithms. NeurIPS, 2012
>
>
>
> **Q4. The advantages of designing hierarchy at the feature level.**
>
> **A4.**
>
> 1. **Carrying Multiple Semantic Cues**
>    A hierarchy built in the raw value space merely propagates numerical value information. In contrast, a feature-level hierarchy embeds additional semantic details (e.g., timestamps, variable identifiers, and sampling density). This enriched representation enables the model to capture not only numerical variations but also temporal dynamics and inter-variable heterogeneity—an essential aspect for effectively modeling IMTS.
> 2. **Efficient Single-time Feature Encoding**
>    With a feature-level hierarchy, a single feature encoding is performed at the outset, and subsequent layers focus solely on dependency extraction and fusion. Constructing the hierarchy at the raw value level would require repeated encoding at every layer, thus incurring higher computational cost.
> 3. **Enhanced Information Transmission and Control**
>    We employ multi-head and multi-time attention mechanisms at the feature level to ensure appropriate weighting and filtering during the aggregation process, thereby mitigating inappropriate information passing. Our ablation experiments (w/o TEAGG) confirm that this design substantially improves high-level feature learning (11.11% on MSE↑ and 5.53% on MAE↑ on four datasets).

---

### Official Review · Reviewer_2e2R · 2025-03-12

**Overall Recommendation:** 4

**Summary:**

The paper introduces a graph-based framework called **Hi-Patch** to handle irregularly sampled multivariate time series. The approach divides the time axis into patches of short intervals, capturing local (fine-grained) temporal patterns for densely sampled variables in each patch. It then progressively aggregates and propagates patch-level information through multiple “inter-patch” graph layers, forming a hierarchical structure that extracts coarse-grained temporal and inter-variable correlations for densely and sparsely sampled variables. This bottom-up, multi-scale pipeline allows for comprehensive feature extraction spanning local to global time scales. Empirical results show that Hi-Patch outperforms a variety of baselines in both classification (area under ROC/PR curves) and forecasting (MSE/MAE) tasks. The method’s key contributions include a patch-based intra-graph for local context, a stack of inter-patch graphs for higher-level features, and a final decoding layer that uses these learned representations for downstream predictions.

## update after rebuttal
The authors covered most of my concerns in the rebuttal, so I kept the original positive rating.

**Claims And Evidence:**

The paper’s primary claims—namely that Hi-Patch achieves state-of-the-art results on irregular multivariate time series forecasting and classification through a novel hierarchical patch-based architecture—are substantiated mainly. The experiments cover multiple datasets of varied nature (e.g., clinical, climate, activity), comparing against both specialized irregular-series baselines and prominent time-series models and typically show consistent improvements in performance metrics (e.g., MSE/MAE, AUROC/AUPRC).

One minor point to note is that while the method’s patch-based hierarchical graph structure is well-tested under moderate sequence lengths, claims about scalability to massive datasets or exceedingly fine-grained irregularities could benefit from deeper empirical exploration. Overall, the evidence presented—including comparative results, ablation studies, and multi-dataset analysis—convincingly supports the paper’s central claims.

**Essential References Not Discussed:**

The paper already cites a range of prior works on patch-based modeling (e.g., PatchTST), continuous-time or irregular methods (e.g., Latent ODEs, GRU-D), and multi-scale networks (e.g., MSGNet). From the perspective of irregular time-series GNNs and multi-scale forecasting, no critical, well-known work appears omitted. The paper’s references collectively offer sufficient background for its primary contributions.

**Experimental Designs Or Analyses:**

The paper’s experimental design appears methodologically sound—particularly its multi-dataset, multi-task approach. The authors follow established splits (e.g., 6:2:2) and compare against well-chosen baselines across domains. They include ablation studies and variance reporting, further bolstering validity. No critical flaws or inconsistencies in the experimental design or analyses were observed.

**Methods And Evaluation Criteria:**

Yes. The authors use benchmark datasets covering domains where irregular multivariate time series naturally arise (e.g., healthcare, climate, motion). These datasets and tasks (forecasting and classification) are appropriate for evaluating the benefits of the proposed hierarchical patch-based method. Comparisons to both general-purpose time-series models and specialized irregular-series frameworks further demonstrate the paper’s relevance to the problem setting and the robustness of its approach.

**Other Comments Or Suggestions:**

Other Comments and Suggestions:
- Overall, the paper is well-written and clearly describes the methodology and experiments.
- Minor typographical errors were observed, such as occasional inconsistencies in notation formatting (e.g., varying use of boldface for vectors) and a few grammatical slips that can be easily corrected during revision.
- A short discussion on potential limitations or failure cases could further balance the presentation.

**Other Strengths And Weaknesses:**

The paper exhibits notable originality by creatively combining patch-based time series segmentation with hierarchical graph neural networks. This approach effectively addresses the challenges of irregular sampling and multi-scale feature extraction. This innovative synthesis bridges ideas from patch representations and continuous-time graph models, filling an essential gap in the literature. The empirical evaluation is comprehensive, spanning diverse datasets and tasks, which underscores the significance of the method across various real-world applications. Additionally, the clarity in the presentation of the model architecture and the experimental setup enhances the paper's overall quality.

On the downside, the increased complexity of the hierarchical design introduces additional hyperparameters and potential computational overhead, which may affect scalability in massive datasets. Furthermore, while the paper demonstrates substantial performance improvements, more discussion on the interpretability of the learned multi-scale features and a deeper analysis of the metamethod's sensitivity to hyperparameter choices could further strengthen the contribution.

**Questions For Authors:**

1. How does the proposed hierarchical framework scale computationally when applied to highly long time series or datasets with significantly higher sampling densities?
2. Can the authors elaborate on any techniques or experiments aimed at interpreting the hierarchical features learned by the model—for instance, identifying which scales or graph connections most strongly influence the final predictions?

**Relation To Broader Scientific Literature:**

The paper extends patch-based approaches like PatchTST—which typically handles regular time series—by leveraging graph-based links to account for the irregular sampling patterns seen in many real-world datasets. This builds on prior GNN research (e.g., tPatchGNN, MTGNN) that models asynchronous inter-variable relationships; however, it is combined with a hierarchical, multi-scale design. The work synthesizes established patching methods in time series analysis with graph architectures that capture local (intra-patch) and global (inter-patch) dependencies, advancing existing literature on irregular time series modeling and multi-scale feature extraction.

**Theoretical Claims:**

The submission does not appear to present formal proofs or a rigorous theoretical framework beyond outlining model architectures and derivations (e.g., attention mechanisms, graph-update formulas). Consequently, there are no explicit proofs to check for correctness. The authors support their modeling choices with empirical evidence rather than theoretical guarantees.

---

> ### Author Rebuttal · Authors · 2025-03-30
>
> # Responses to Reviewer 2e2R
>
> **W1 & Q1. The increased complexity of the hierarchical design may affect scalability in massive datasets. How does it scale computationally when applied to highly long time series or datasets with significantly higher sampling densities?**
>
> **A1.**
>
> 1. **Complexity in Sparse IMTS**
>
>    As detailed in Appendix G.1, **the primary computational cost of our method stems from the intra-patch graph layer**, whose complexity is quadratic in the number of observation points (N) within each patch rather than hierarchical design.
>
> 2. **Future Scalability on Dense Data**
>
>    In scenarios involving highly long time series or datasets with much higher sampling densities, we can readily adopt existing scalability techniques to mitigate the overhead of the intra-patch graph.
>
>    For instance, **edge pruning**—where each node connects only to its nearest k observation points—can reduce the complexity to **O(kN)**. Alternatively, one could improve the attention mechanism (e.g., by incorporating the **ProbSparse Self-Attention** from Informer, which reduces complexity to **O(N log N))**.
>
>    In these cases, the trade-off between a slight reduction in feature extraction capability and improved scalability would be justified. This extension is part of our planned future work; the current paper focuses on the initial proposal of a multi-scale framework for sparse IMTS. In the revised version, we will add a Limitations section to discuss scalability aspects.
>
> ------
>
> **W2 & Q2. More discussions or experiments on the interpretability of the learned hierarchical multi-scale features.**
>
> **A2.** Taking the USHCN dataset as an example—comprising climate data from 1,218 centers across the United States for five variables (precipitation, snowfall, snow depth, maximum temperature, and minimum temperature)—we use the first two years (24 months) as historical input to predict future changes.
>
> 1. **Effect of Different Scales on Final Prediction Results**
>
>    As illustrated in Figure 3(a) in Section 5.4, increasing the number of scales from 1 to 5 (by incorporating scales of 12, 6, 3, and 1.5 months) results in a continuous decrease in MSE, demonstrating the benefit of multi-scale features. Notably, when a 0.75-month scale is added (resulting in 6 scales), the MSE increases, indicating that this extra scale introduces redundancy rather than additional useful information.
>
>    **This experiment not only identifies the scales that are beneficial for prediction, but also reveals which scales most strongly influence the final predictions through the slope changes.** For instance, the largest drop occurs when the 6-month scale is introduced, highlighting it is the most important scale for weather prediction.
>
> 2. **Discussion on the Interpretability of Learned Multi-Scale Features**
>
>    As shown in Figure 5(a) in Section 5.5, our model achieves optimal performance with a patch size of 1.5 months, which enables feature extraction at scales of 1.5, 3, 6, and 12 months. **These scales correspond well with key climatological cycles and are largely interpretable.** Figure 12 in Appendix K provides visualizations of the time series at these scales based on our patching approach:
>
>    - **Scale 1 (Original Scale):** The original scale view retains the full sequence data, exhibiting high local variability.
>    - **Scale 2 (1.5-Month Aggregation):** By aggregating data every 1.5 months, the resulting view smooths short-term fluctuations, highlighting mid-term climate trends.
>    - **Scale 3 (3-Month Cycle):** Aggregating every two observations from Scale 2 yields a 3-month cycle view. Seasonal trends become more evident in this view, with red and purple temperature curves show an upward, downward, upward and downward trend, broadly corresponding to the four seasons.
>    - **Scale 4 (6-Month Cycle):** Further aggregation at 6-month intervals reveals monsoonal climate trends, where the red and purple temperature curves first rise and then fall, reflecting the impact of summer and winter monsoons.
>    - **Scale 5 (12-Month Cycle):** At a 12-month aggregation, the view reveals annual macro-climate trends, such as overall increases, decreases, or stabilization in climate patterns.
>
>    We will include this analysis in the appendix in future revisions.
>
> ------
>
> **Comment1. Minor typographical errors and grammatical slips.**
>
> **A3.** Thank you for your careful review. We will thoroughly proofread and correct these issues in the revised version of the paper.
>
> ------
>
> **Comment2. A short discussion on potential limitations or failure cases.**
>
> **A4.** We will incorporate a discussion on the scalability limitations of Hi-Patch in light of our response to W1 & Q1.

---

> > ### Comment · Reviewer_2e2R · 2025-04-04
> >
> > The author's response somewhat alleviated my doubts, so I retained the original positive rating.

---

> > > ### Author Response · Authors · 2025-04-05
> > >
> > > Thank you for maintaining your positive rating. We’re glad to have alleviated your concerns and would be happy to clarify any further questions you may have.

---

### Official Review · Reviewer_ZR6y · 2025-03-15

**Overall Recommendation:** 3

**Summary:**

This paper introduces a Hierarchical Patch Graph Neural Network (Hi-Patch) for Irregular Multivariate Time Series (IMTS) modeling, where variables have distinct sampling rates and exhibit multi-scale dependencies. Existing multi-scale analysis methods struggle with IMTS due to their assumption of regular sampling, making them ineffective in handling asynchronous and mixed-granularity data. Hi-Patch addresses this by first encoding each observation as a node, capturing local temporal and inter-variable dependencies through an intra-patch graph layer, and then progressively aggregating these nodes into higher-level patch representations using inter-patch graph layers. This hierarchical structure enables multi-scale feature extraction, preserving both fine-grained and coarse-grained patterns across different variable scales. The model is evaluated on forecasting and classification tasks across eight datasets, outperforming state-of-the-art methods.

**Claims And Evidence:**

yes

**Essential References Not Discussed:**

The key related works are included.

**Experimental Designs Or Analyses:**

yes

**Methods And Evaluation Criteria:**

yes

**Other Comments Or Suggestions:**

NA

**Other Strengths And Weaknesses:**

Strengths:
1. The studied classification and forecasting problems of Irregular Multivariate Time Series (IMTS) have significant applications in various crucial domains, such as healthcare, finance, climate science, and astronomy.
2. The proposed Hi-Patch aims to address several unique challenges posed by IMTS modeling, including irregularity, asynchrony, and mixed granularity. These challenges may not be present and addressed in the widely studied Regular Multivariate Time Series.
3. The experimental results exhibit promising performance in IMTS forecasting task.

Weaknesses:
1. The Hi-Patch model appears computationally expensive due to its use of graph attention operations on fully connected graphs of all observation nodes across variables. This may limit its scalability for large-scale variables with dense observations.
2. The performance improvement in the classification task seems marginal. Additionally, the PhysioNet and P12 datasets both originate from The PhysioNet/Computing in Cardiology Challenge 2012 [1], making them somewhat redundant.
3. As a crucial driving factor for this study, the motivation of studying multi-scale information in IMTS should be further justified. Why is it an important factor in IMTS modeling?

[1] https://physionet.org/content/challenge-2012/1.0.0/.

**Questions For Authors:**

see weakness

**Relation To Broader Scientific Literature:**

The proposed Hi-Patch might be applied to IMTS modeling across various domains.

**Theoretical Claims:**

No proofs in the paper

---

> ### Author Rebuttal · Authors · 2025-03-30
>
> # Responses to Reviewer ZR6y
>
> **W1. Hi-Patch appears computationally expensive, which may limit its scalability.**
>
> **A1.** We address this concern from both sparse and dense data perspectives.
>
> 1. **Trade-off in Sparse IMTS**
>
>    As detailed in Appendix G.1, the primary computational cost of our method arises from the intra-patch graph layer, whose complexity scales quadratically with the number of observation points (N) per patch.
>
>    Our focus is on irregular multivariate time series, which are typically characterized by extremely sparse sampling. As shown in Table 4, all our datasets exhibit a missing rate exceeding **75%**.
>
>    In these scenarios, the computational cost decreases sharply due to the limited number of available data points. We contend that the additional overhead is justified by the improved extraction of scarce data patterns. Moreover, our empirical study in Appendix G.2 indicates that our overhead is moderate—**ranking 5th out of 9 methods**—and remains within an acceptable range.
>
> 2. **Future Scalability on Dense Data**
>
>    Should the need arise to extend our approach to large-scale dense datasets, existing scalability techniques can be integrated to reduce the overhead of the intra-patch graph.
>
>    For instance, **edge pruning**—such as connecting each node only to its nearest k observation points—can reduce the complexity to **O(kN)**, or alternatively, leveraging improved attention mechanisms like the **ProbSparse Self-Attention** from Informer can lower the complexity to **O(N log N)**.
>
>    In these cases, the trade-off between a slight reduction in feature extraction and enhanced scalability would be worthwhile. This extension is among our planned future works. For the current paper, our core contribution remains the introduction of a multi-scale framework for sparse IMTS. We will also add a Limitations section discussing model scalability.
>
> ------
>
> **W2. The performance improvement in the classification task seems marginal. The PhysioNet and P12 datasets are redundant.**
>
> **A2.**
>
> 1. **Classification Performance Improvement**
>
>    Although our method only shows marginal improvements over the best baseline on each individual dataset, it is important to note that **no single baseline consistently outperforms the others across all datasets**. The table below presents the average AUROC and AUPRC for our method and the three most competitive baselines across four datasets. **Our method achieves an average AUROC improvement of 1% and an average AUPRC increase of 2.7%**, which is noteworthy given that the AUROC values are already near 90%.
>
>    |            | Avg AUROC  | Avg AUPRC  |
>    | ---------- | ---------- | ---------- |
>    | IP-Net     | 86.2       | 52.0 (2nd) |
>    | StraTS     | 86.7 (2nd) | 51.1       |
>    | Warpformer | 86.3       | 50.3       |
>    | Ours       | 87.7 (1st) | 54.7 (1st) |
>
> 2. **PhysioNet and P12 Datasets**
>
>    The P12 dataset comprises three subsets—set-a, set-b, and set-c—each containing 4,000 samples (totaling 12,000 samples). Previous studies [1, 2] have utilized only the set-a subset (namely PhysioNet dataset in our paper), whereas others [3, 4] employed the full P12 dataset.
>
>    **By evaluating both configurations, we aim to test the robustness of our method with respect to sample size and distribution.** The consistent performance across these configurations further validates the stability of our approach.
>
>    [1]. Multi-Time Attention Networks for Irregularly Sampled Time Series, ICLR, 2021
>
>     [2]. Warpformer: A Multi-scale Modeling Approach for Irregular Clinical Time Series, KDD, 2023
>
>     [3]. Graph-Guided Network for Irregularly Sampled Multivariate Time Series, ICLR, 2022
>
>     [4]. Time Series as Images: Vision Transformer for Irregularly Sampled Time Series, NeurIPS, 2023
>
> ------
>
> **W3. The motivation for studying multi-scale information in IMTS.**
>
> **A3.**
>
> 1. **Foundational Evidence**
>
>    Prior studies—such as those on Pathformer, TimeMixer, and MSGNet—have demonstrated that multi-scale information is crucial for general time series analysis. Although irregular multivariate time series (IMTS) are characterized by uneven sampling intervals, they inherently retain multi-scale properties.
>
> 2. **Practical Example**
>
>    For instance, the USHCN weather dataset exhibits monthly, quarterly, and yearly patterns (see Figure 12). Each temporal scale shows distinct patterns that are essential for accurate prediction. This example underlines the importance of multi-scale information in IMTS.
>
> 3. **Prior Empirical Validation**
>
>    Empirical evidence from previous work Warpformer [2], supports the benefits of incorporating multi-scale information into IMTS modeling. While Warpformer has its own limitations, its success further motivates our deeper exploration into a multi-scale modeling approach for IMTS.
>
> Collectively, these points underscore the significance of multi-scale information in IMTS modeling and reinforce the motivation behind our work.

---

> > ### Comment · Reviewer_ZR6y · 2025-04-03
> >
> > Thanks for the clarification, I will raise my score.

---

> > > ### Author Response · Authors · 2025-04-04
> > >
> > > Thank you for your positive feedback and for taking the time to carefully review our responses. We deeply appreciate your thoughtful evaluation and support.

---

### Decision · Program_Chairs · 2025-05-01

**Decision:**

Accept (poster)

**Comment:**

This paper introduces Hi-Patch, a hierarchical patch-based GNN framework for modeling Irregular Multivariate Time Series (IMTS). The approach effectively captures multi-scale temporal and inter-variable dependencies through intra- and inter-patch graph layers. The reviewers appreciated the paper's novelty, solid empirical results across diverse datasets, and relevance to real-world IMTS applications. Concerns were raised about scalability, clarity of multi-scale motivation, and interpretability of learned features. The authors provided thorough and satisfactory rebuttals, addressing computational complexity, validating the importance of hierarchy via ablations, and adding insights into scale interpretability and practical impact. Multiple reviewers raised or confirmed their scores post-rebuttal.